# Grasp2Grasp: Vision-Based Dexterous Grasp Translation via Schrödinger Bridges

**Tao Zhong[1], Jonah Buchanan[†2,3], Christine Allen-Blanchette[1]**
[1]Princeton University,   [2]San Jose State University,   [3]Lockheed Martin Corporation
{tzhong, ca15}@princeton.edu   jonah.buchanan808@gmail.com

## Abstract

We propose a new approach to vision-based dexterous grasp translation, which aims to transfer grasp intent across robotic hands with differing morphologies. Given a visual observation of a source hand grasping an object, our goal is to synthesize a functionally equivalent grasp for a target hand without requiring paired demonstrations or hand-specific simulations. We frame this problem as a stochastic transport between grasp distributions using the Schrödinger Bridge formalism. Our method learns to map between source and target latent grasp spaces via score and flow matching, conditioned on visual observations. To guide this translation, we introduce physics-informed cost functions that encode alignment in base pose, contact maps, wrench space, and manipulability. Experiments across diverse hand-object pairs demonstrate that our approach generates stable, physically grounded grasps with strong generalization. This work enables semantic grasp transfer for heterogeneous manipulators and bridges vision-based grasping with probabilistic generative modeling. Additional details at grasp2grasp.github.io.

## 1 Introduction

Robotic grasping remains a central challenge in autonomous manipulation, especially in the context of dexterous, multi-fingered hands. Unlike parallel-jaw grippers or simple suction devices, dexterous hands offer significantly more control and flexibility but also bring high-dimensional configuration spaces [3, 18] and non-trivial contact dynamics [65]. These properties pose difficulties [53] in data collection, modeling, and learning robust grasp strategies. While recent data-driven works [9, 46, 52, 91, 100, 101] have shown success in generating feasible grasps from large-scale datasets [32, 46, 87, 94], a key limitation persists: models typically operate in a hand-specific setting. This lack of generalization requires retraining or significant fine-tuning to adapt to new robotic hands [41], which poses a practical bottleneck in the rapidly evolving landscape of robotic hardware and limits the applicability of learned policies in real-world systems [97].

Therefore, it is desirable to reuse the grasp knowledge acquired from one hand to infer meaningful grasp strategies for another. However, the morphological differences between hands mean that naive transfer, such as copying joint angles or end-effector poses, might lead to physically invalid or unstable grasps [2, 58, 68]. A more fundamental challenge is the acquisition of paired demonstration data, where a grasp for a source hand is mapped to a functionally equivalent grasp for a target. Manually creating or algorithmically discovering such pairings at scale is extremely difficult. This motivates the need for a framework that can learn to translate the intent of a grasp from a source hand to a target hand using only unpaired, hand-specific grasp datasets, leveraging the underlying object geometry and physics to bridge the morphological gap.

---

[†]Buchanan contributed to this work in the capacity as an independent researcher. The views expressed (or the conclusions reached) are their own and do not necessarily represent the views of Lockheed Martin Corporation.

39th Conference on Neural Information Processing Systems (NeurIPS 2025).

This paper aims to address the problem of vision-based dexterous grasp translation. As shown in Fig. 1, given a visual observation of a source robotic hand grasping an object, we aim to generate a plausible grasp for a target hand with a different morphology. This setting arises naturally in scenarios such as human-to-robot imitation [71, 96], hardware substitution in manipulation pipelines [30], and learning from heterogeneous demonstrations [38]. Although reminiscent of teleoperation [24, 37] or behavior cloning [16, 97], our formulation diverges in a critical way: the goal is not to replicate the precise joint configuration or trajectory of the source hand.

Instead, we seek to preserve the functional intent of the grasp, expressed through physical quantities such as contact locations, grasp wrenches, or stability measures, while accounting for the morphological differences between the source and target manipulators.

To this end, we cast the grasp translation task as a probabilistic transport problem between the distribution of grasps executed by the source hand and those executable by the target hand. We leverage the formalism of the Schrödinger Bridge (SB) [45, 77], which models the most likely stochastic process interpolating between two distributions under a reference dynamics. This perspective allows us to frame grasp translation as learning a distributional flow from observed source grasps to compatible target grasps, guided by a meaningful notion of grasp similarity and grounded in physics.

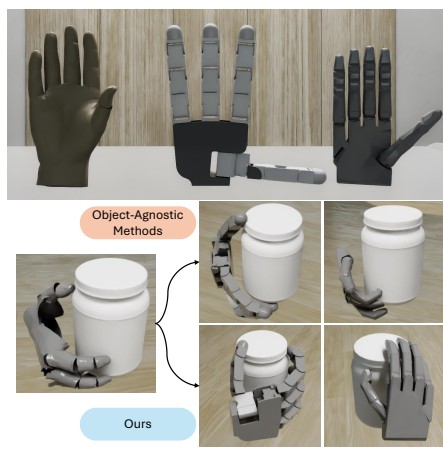

Figure 1: (*top*) Comparison of morphology and scale across hands. (*bottom*) Object-agnostic methods often miss fine-grained contacts, leading to invalid grasps. Our method produces contact-consistent, stable grasps across diverse hand morphologies.

Our approach builds on a growing body of work [15, 21, 47, 79, 84, 90, 93] that frames generative modeling as a Schrödinger Bridge problem, which enables probabilistic transport between arbitrary data distributions under stochastic dynamics. In particular, recent developments have demonstrated how score-based [21, 93] and flow-based [22, 84, 85] objectives can be leveraged to learn such bridges without the need for simulating full stochastic processes, offering scalable and flexible tools for distribution alignment. We adapt and extend these ideas to the domain of dexterous grasp translation, where the source and target distributions represent physically meaningful grasps from different robotic hands. To this end, we develop a latent-space learning pipeline for distribution-level grasp translation conditioned on vision-based observations of the source hand. Within this framework, we design custom ground cost functions that capture task-relevant physical properties, such as contact patterns and grasp stability, rather than relying on generic Euclidean or geodesic distances. This allows us to generate grasps that are not only kinematically feasible for the target hand but also functionally aligned with the intent of the source grasp.

We summarize our contributions as follows:

- We formulate vision-based dexterous grasp translation as a Schrödinger Bridge problem between source and target grasp distributions, capturing the stochastic nature of plausible grasps.

- We introduce novel ground cost functions tailored to the grasping domain, enabling semantic and physically-informed translation across different hand morphologies.

- We develop a latent-space learning pipeline based on score and flow matching, enabling simulation-free, distribution-level training on unpaired visual grasp observations.

- We evaluate our method across diverse hand-object combinations, demonstrating that our approach generates functionally equivalent and physically robust grasps with strong generalization capabilities.

## 2  Related Work

**Dexterous Grasp Generation** has been a long-standing topic in robotics, aimed at enabling robotic hands with multiple degrees of freedom (DoFs) to securely and effectively manipulate diverse objects. Early approaches [6, 8, 23, 29, 62, 67] relied on analytical models of hand kinematics and contact

mechanics [7, 60] to compute force-closure [49, 62, 64, 67, 87] or form-closure [59, 86] grasps, which are often constrained by assumptions of perfect information and simple object geometries [7, 60]. The advent of vision-based data-driven techniques has led to significant advances [66, 99], particularly through direct regression [48, 76] or generative modeling techniques [33, 34, 42] such as VAEs [41, 46, 61], GANs [19, 54, 55, 61], and diffusion models [36, 52, 88, 95]. These generative approaches enable the synthesis of diverse grasp configurations conditioned on visual inputs such as RGB-D images [39, 40] or segmented point clouds [36, 52, 95], allowing for more flexible and scalable grasp generation. For example, GenDexGrasp [46] introduces a hand-agnostic grasping algorithm trained on the MultiDex dataset [46], which leverages contact maps as intermediate representations to efficiently generate diverse and plausible grasping poses transferable among various multi-fingered robotic hands. Similarly, UGG [52] presents a unified diffusion-based dexterous grasp generation model that operates within object point cloud and hand parameter spaces. While these generative approaches have advanced the field, they often remain tailored to specific robotic hands and lack mechanisms for transferring grasp knowledge across different hand morphologies. Our work addresses these limitations by developing a grasp translation framework that operates at the distributional level, enabling the reuse of grasp knowledge across different robotic platforms through a physically meaningful transport mechanism.

**Grasp Transfer** is closely related to teleoperation [72] and imitation learning [4], where the objective is to map motions or actions from a source agent (e.g., a human or a robot) to a target robot. Teleoperation methods [24, 37, 70, 78, 80, 104] often aim for one-to-one replication of joint trajectories or end-effector poses, which is generally effective when the source and target share similar kinematics. However, such approaches struggle when transferring grasps between manipulators with significantly different morphologies, as direct replication can lead to infeasible or suboptimal grasps [2, 58, 68]. To address this morphological gap, retargeting methods have been explored. For instance, works like [14, 35, 72] explore positional optimization for key points, while CrossDex [98] uses a neural network trained with data generated from [72] for retargeting. These approaches, however, often rely on pre-defined link-to-link correspondences or human-annotated templates, which can limit their generalizability. More recent work, such as RobotFingerPrint [41], addresses this challenge by mapping grippers into a shared Unified Gripper Coordinate Space (UGCS) based on spherical coordinates to facilitate grasp transfer. While promising, this approach requires simulation-based preprocessing and detailed gripper models, which impose a strong prior that limits flexibility. Moreover, the UGCS approach focuses on point correspondences and is thus object-agnostic, rather than explicitly preserving functional grasp intent. In contrast, our formulation aims to translate grasps in a distributional sense grounded in physical properties. It requires no hand-specific simulation preprocessing, which enables broader applicability and more semantically meaningful grasp transfer.

**Optimal Transport (OT) and Schrödinger Bridges (SB)** have recently emerged as powerful frameworks in the machine learning community for modeling distributional transformations. OT [69, 75, 81, 92] provides a principled way to compute mappings between probability distributions that minimize a cost function, typically grounded in geometric distances. Aligning distributions is a central challenge in tasks like domain adaptation [17, 102], and classical OT has been successfully applied for this purpose [20, 27], as well as for other applications like shape matching [25, 44, 83]. However, it often requires solving computationally intensive optimization problems [63, 81]. Schrödinger Bridges [45, 77] extend OT by introducing a stochastic reference process, which enables more flexible and entropically regularized transport. This has proven particularly useful in generative modeling [15, 21, 47, 79, 84, 90, 93], where SBs offer a probabilistic interpolation between data distributions. Recent advances in simulation-free training methods [50, 84, 85] have made these techniques more scalable by bypassing the need to simulate stochastic processes during training. In robotics, these frameworks have been explored for motion planning [43], distribution alignment [82], and dynamical system modeling [10]. However, their application to grasp synthesis and translation remains underexplored. Our work leverages SBs not just for smooth transport between distributions, but for preserving task-specific physical properties of grasps across different hand morphologies, pushing the boundaries of OT/SB utility in robotic manipulation.

## 3 Preliminaries

### 3.1 Schrödinger Bridge Formulation

The Schrödinger Bridge (SB) problem [45, 77] provides a principled framework for defining the most likely stochastic evolution between two probability distributions, a source $q_0(x)$ and a target

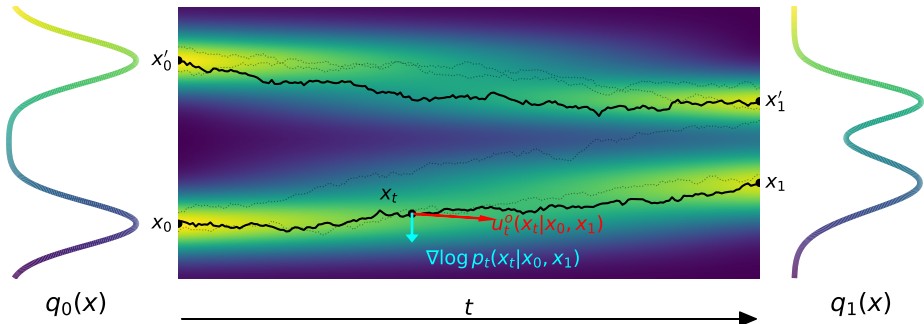

$q_0(x)$        $t$        $q_1(x)$

Figure 2: **Illustration of the Schrödinger Bridge Models.** The Schrödinger Bridge process transports samples from a source distribution ($q_0$) to a target ($q_1$). At an intermediate point $x_t$ on a Brownian bridge trajectory between coupled samples $(x_0, x_1)$, our model learns the conditional flow drift $u_t^\circ$ that drives transport and the conditional score $\nabla \log p_t$ that corrects the path.

$q_1(x)$, under a reference process. The optimal process can be found by weighting paths according to an entropically regularized optimal transport (OT) plan, $\pi_\varepsilon^*$. This plan seeks a coupling between the distributions that minimizes a ground cost $d(\cdot, \cdot)$ plus a regularization term:

$$\pi_\varepsilon^* = \underset{\pi \in U(q_0, q_1)}{\arg\min} \int d(x_0, x_1)^2 d\pi(x_0, x_1) + \varepsilon \mathrm{KL}(\pi || q_0 \otimes q_1), \tag{1}$$

where $U(q_0, q_1)$ is the set of all couplings between $q_0$ and $q_1$. This connection allows the complex dynamic optimization to be grounded in a more tractable static transport plan. We refer readers to Appendix A for a detailed formulation.

### 3.2 Score-based and Flow-Matching Methods

We model the stochastic dynamics of the SB with an Itô SDE of the form:

$$dx_t = u_t(x_t)dt + g(t)dw_t, \tag{2}$$

where $u_t(x)$ is a drift vector field and $w_t$ is standard Brownian motion. This process induces time marginals $p_t(x)$ that can also be described by an equivalent deterministic probability flow ODE:

$$dx_t = \underbrace{\left( u_t(x_t) - \frac{g(t)^2}{2} \nabla \log p_t(x_t) \right)}_{u_t^\circ(x)} dt, \tag{3}$$

Following the [SF]$^2$M framework [84], we learn the stochastic dynamics by training neural networks $v_\theta(t, x)$ and $s_\theta(t, x)$ to approximate the ODE drift ($u_t^\circ$) and the score term ($\nabla \log p_t(x)$), respectively. The training targets are derived from conditional Brownian bridge paths between pairs of samples $(x_0, x_1)$ drawn from the entropic OT plan $\pi_\varepsilon^*$ (Eq. 1). A visualization of these conditional targets is provided in Fig. 2.

The resulting conditional loss function combines flow and score matching terms:

$$\mathcal{L}_{[\mathrm{SF}]^2\mathrm{M}}(\theta) = \underset{\substack{t \sim \mathcal{U}(0,1) \\ x \sim p_t(x|x_0,x_1) \\ (x_0,x_1) \sim \pi_\varepsilon^*}}{\mathbb{E}} \left[ \| v_\theta(t, x) - u_t^\circ(x|x_0, x_1) \|^2 + \lambda(t)^2 \| s_\theta(t, x) - \nabla \log p_t(x|x_0, x_1) \|^2 \right],$$

$$\tag{4}$$

where $u_t^\circ(\cdot)$ and $\nabla \log p_t(\cdot)$ are the conditional ODE drift and score targets, and $\lambda(t)$ is a weighting schedule. The full derivation of the training targets and weighting schedule is provided in Appendix A.

### 3.3 Problem Definition

We formalize the vision-based dexterous grasp translation problem as a stochastic transport task between grasp distributions associated with different robotic hands. Let $\mathcal{G}_{\mathrm{source}} \subset \mathbb{R}^n$ and $\mathcal{G}_{\mathrm{target}} \subset \mathbb{R}^m$ denote the source and target hand configuration spaces with $n$ and $m$ degrees of freedom, including the pose of the hand base in $SE(3)$, and let $\mathcal{O}$ represent the space of grasp observations. Given a segmented visual observation $(o_{\mathrm{obj}}, o_{\mathrm{hand}}) \in \mathcal{O}$ depicting the source hand with configuration

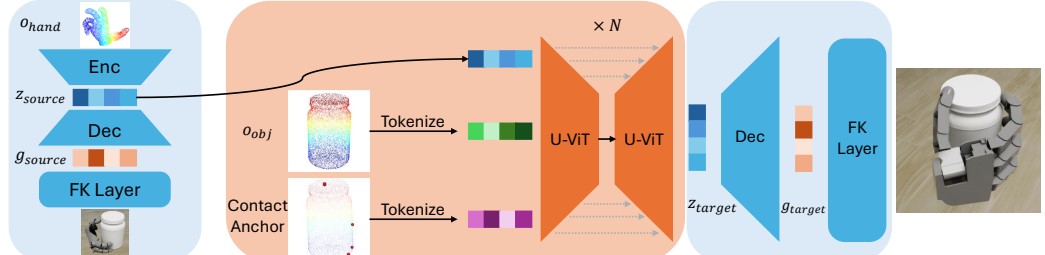

Figure 3: **Architecture overview.** Blue modules correspond to stage 1: the source hand observation is encoded via a VAE. Orange modules correspond to stage 2: the latent is translated using a U-ViT Schrödinger Bridge model conditioned on object shape and contact anchors. The translated latent is decoded to produce the target hand grasp.

$g_{\text{source}} \in \mathcal{G}_{\text{source}}$ grasping an object, our goal is to generate a corresponding grasp configuration $g_{\text{target}} \in \mathcal{G}_{\text{target}}$ for the target hand such that certain physical grasp properties are preserved.

We assume that grasps from the source hand follow a distribution[1] $q_0(g_{\text{source}}|o_{\text{obj}})$, while feasible grasps for the target hand follow $q_1(g_{\text{target}}|o_{\text{obj}})$, both conditioned on the same object observation. These distributions capture the variability and physical plausibility of grasps within each hand's morphology. The problem is to learn a mapping between $q_0$ and $q_1$ that respects both the structure of the source grasp and the constraints of the target hand.

To achieve this, we define a stochastic process $\{g^t_{\text{target}}\}_{t \in [0,1]}$ interpolating between $q_0$ and $q_1$, governed by a Schrödinger Bridge with a suitable reference dynamics. The transport should minimize the deviation from a reference process (e.g., Brownian motion) while aligning the marginal distributions at $t = 0$ and $t = 1$. Crucially, a common choice [84, 85] in related works is to define the transport cost $d(\cdot, \cdot)$ in terms of Euclidean or geodesic distances, which primarily capture geometric dissimilarities. However, such measures are insufficient in our context, as they do not reflect the functional equivalence of grasps. Therefore, we also require a physically grounded notion of cost that considers the grasp's effect on the object, motivating the design of more expressive transport mechanisms aligned with the task's semantics.

The translation task thus reduces to learning a generative model for $q_1(g_{\text{target}}|o_{\text{obj}})$ that aligns with $q_0(g_{\text{source}}|o_{\text{obj}})$ under the SB formulation, leveraging both vision-based cues and physical constraints.

## 4 Methodology

### 4.1 Latent Score and Flow Matching

We now apply the general Schrödinger Bridge framework from Sec. 3 to our specific problem of grasp translation. The abstract distributions $q_0$ and $q_1$ become the latent distributions of source and target grasps, respectively. The transport process between them is learned by optimizing the [SF]$^2$M objective from Eq. 4 in a latent space. We approach grasp translation as stochastic transport in a learned latent space, where the high-dimensional grasp configurations for source and target hands are first embedded into a shared latent representation. This is achieved through a two-stage pipeline, as illustrated in Fig. 3. In the first stage, we train a variational autoencoder (VAE) [42] that encodes segmented visual observations of the source hand grasping an object into a latent variable $z \in \mathcal{Z}$. The VAE consists of an encoder mapping $\mathcal{O} \to \mathcal{Z}$ and a decoder mapping $\mathcal{Z} \to \mathcal{G}$. The decoded grasp configuration $\hat{g} = \text{Dec}(z)$ is passed through a differentiable kinematic layer $\text{FK} : \mathcal{G} \to \mathbb{R}^{3N}$, yielding the 3D mesh vertices of the hand $\hat{v} = \text{FK}(\hat{g})$. The VAE is optimized with a combined loss with a small KL term:

$$\mathcal{L}_{VAE} = \mathbb{E}_{o \sim \mathcal{O}}[\|\hat{g} - g\|^2 + \alpha \|\hat{v} - \text{FK}(g)\|^2] + \beta \text{KL}\left(q_{\text{Enc}}(z|o)\|\mathcal{N}(0, I)\right), \qquad (5)$$

where $g$ denotes the ground truth grasp configuration and $\alpha, \beta$ are weighting coefficients. This training encourages the latent space $\mathcal{Z}$ to encode semantically meaningful and physically grounded grasp features that generalize across hands.

---

[1]Here, $g_{\text{source}}$ is implicitly defined by $o_{\text{hand}}$.

**Algorithm 1** VAE Training

**Require:** a dataset of $(o, g)$, initial network Enc, Dec, FK layer
1: **for all** $(o, g)$ in dataset **do**
2:      $z \sim \text{Enc}(o)$
3:      $\hat{g} \leftarrow \text{Dec}(z)$
4:      $\hat{v} \leftarrow \text{FK}(\hat{g})$
5:      Update $\mathcal{L}_{\text{VAE}}$ with Eq. 5
6:      Update Enc, Dec
7: **end for**
8: **return** Enc, Dec

**Algorithm 2** Latent SB

**Require:** Trained Enc, source and target dataset, OTPlan, initial $v_\theta, s_\theta$
1: **while** Training **do**
2:      Sample $o_s$, $o_t$ from dataset
3:      $z_s \leftarrow \text{Enc}(o_s)$
4:      $z_t \leftarrow \text{Enc}(o_t)$
5:      $\pi_\varepsilon^* \leftarrow \text{OTPlan}(z_s, z_t)$
6:      $(z_0, z_1) \sim \pi_\varepsilon^*$
7:      $t \sim \mathcal{U}(0, 1)$
8:      $z^t \sim p_t(z \mid z_0, z_1)$
9:      Update $\mathcal{L}$ with Eq. 4
10:      Update $v_\theta, s_\theta$
11: **end while**
12: **return** $v_\theta, s_\theta$

**Algorithm 3** Inference

**Require:** Trained Enc, Dec, $v_\theta$, $s_\theta$, test observation $o_s$
1: $z_0 \leftarrow \text{Enc}(o_s)$
2: $z_1 \sim p_1(z|o_s)$ using $v_\theta, s_\theta$
3: $\hat{g}_t \leftarrow \text{Dec}(z_1)$
4: **return** $\hat{g}_t$

**Algorithm 1**: Training and inference procedures. **Left:** VAE training. **Center:** Simulation-free Schrödinger Bridge training in latent space. **Right:** Inference by latent evolution.

In the second stage, we model a Schrödinger Bridge between the latent source and target distributions, $q_0(z_{\text{source}}|o_{\text{obj}})$ and $q_1(z_{\text{target}}|o_{\text{obj}})$, with a simulation-free framework. The stochastic path $z^t \in \mathcal{Z}$, $t \in [0, 1]$ is governed by the SDE in Eq. 2 with marginals $p_t(z)$ and corresponding deterministic flow ODE as in Eq. 3.

As introduced in Sec. 3.2, we train neural networks $v_\theta(t, z)$ and $s_\theta(t, z)$ to approximate the conditional flow $u_t^o(z|z_0, z_1)$ and score $\nabla \log p_t(z|z_0, z_1)$, respectively. The training is performed using conditional Brownian bridge samples $z^t \sim p_t(z|z_0, z_1)$ where $(z_0, z_1) \sim \pi_\varepsilon^*$, which is the entropic OT plan defined in Eq. 1 between $q_0$ and $q_1$. The training objective is then defined as Eq. 4.

During inference, given a visual observation of a grasp with the source hand $(o_{\text{obj}}, o_{\text{hand}})$, we sample a latent trajectory starting from a source encoding $z_0 = \text{Enc}(o_{\text{hand}}) \sim q_0(z_{\text{source}}|o_{\text{obj}})$ and evolve it forward using the learned SDE dynamics to obtain a translated grasp latent code $z_1 \sim q_1(z_{\text{target}}|o_{\text{obj}})$. This code is decoded to produce the target hand configuration $g_{\text{target}} = \text{Dec}(z_1)$, which is then used to actuate the target hand. The training and inference processes are detailed in Alg. 1, 2, and 3.

A central challenge in learning this SB is the absence of paired samples between source and target hand configurations. Without a principled coupling, arbitrary samples from $q_0$ and $q_1$ may lead to latent translations that are uninformative or misaligned. This motivates the next section, where we introduce physically grounded ground costs for entropic OT to induce meaningful correspondences between source and target grasps.

### 4.2 Grasp-specific OT Cost Design

To guide the Schrödinger Bridge, we must first compute an entropic OT plan $\pi_\varepsilon^*$ (Eq. 1), which is defined by a ground cost $d(\cdot, \cdot)$. We design several physics-informed costs for this purpose. For a given cost, we compute the OT plan between minibatches of source and target latent samples. We then draw pairs $(z_0, z_1) \sim \pi_\varepsilon^*$ and use these pairs to construct the conditional training targets for the SB model as described in Sec. 3.2 and used in the loss function of Eq. 4. This allows us to enable meaningful grasp translation between different hand morphologies by guiding the process with a ground cost that reflects task-relevant physical properties rather than naive geometric alignment. Conventional entropic OT approaches [84, 85] typically define the ground cost $d(\cdot, \cdot)$ using Euclidean or geodesic distances in either configuration or latent space. However, such metrics fail to capture the functional equivalence of grasps, particularly when source and target hands differ significantly in kinematics and contact modalities. We propose a set of physics-informed cost functions that better encode grasp quality and intent.

**Base Pose Cost.** We define a cost $d_{\text{pose}}$ on the global wrist pose in $SE(3)$ between the source and target hands. The pose is represented as a concatenation of the 3D base position and a 6D continuous representation for the rotation [103] to ensure smoothness. The 6D vector is first converted to a $3 \times 3$ rotation matrix $R \in SO(3)$. The cost then combines the squared L2 norm for translation and the squared Frobenius norm for rotation:

$$d_{\text{pose}} = \|h_{\text{source}} - h_{\text{target}}\|_2^2 + \|R_{\text{source}} - R_{\text{target}}\|_F^2, \tag{6}$$

where $h \in \mathbb{R}^3$ is the translation vector and $R \in SO(3)$ is the rotation matrix of the base pose, extracted from $g_{\text{source}}$ and $g_{\text{target}}$, respectively. This cost encourages coarse alignment of the grasps in workspace coordinates.

**Contact Map Similarity.** Each grasp is associated with a contact map indicating the location of contacts on the object surface. We represent these contact maps as 3D point clouds and compute the bidirectional Chamfer distance between the source and target maps $C_{\text{source}}$, $C_{\text{target}}$:

$$d_{\text{contact}} = \text{Chamfer}(C_{\text{source}}, C_{\text{target}}) = \sum_{x \in C_{\text{source}}} \min_{y \in C_{\text{target}}} \|x - y\|^2 + \sum_{y \in C_{\text{target}}} \min_{x \in C_{\text{source}}} \|y - x\|^2. \tag{7}$$

This encourages preservation of local interaction geometry across morphologically distinct hands.

**Grasp Wrench Space Overlap.** The grasp wrench space (GWS) [29] characterizes the set of wrenches (forces and torques) that can be exerted on an object by a grasp. For each grasp, we compute a set of wrench vectors from contact locations and normals, then take the convex hull to approximate the GWS in $\mathbb{R}^6$ to form the grasp wrench hull (GWH). We define the cost based on the intersection-over-union (IoU) of GWH volumes between source and target hulls:

$$d_{\text{wrench}} = 1 - \frac{\text{Vol}\left(\text{Hull}(GWS_{\text{source}}) \cap \text{Hull}(GWS_{\text{target}})\right)}{\text{Vol}\left(\text{Hull}(GWS_{\text{source}}) \cup \text{Hull}(GWS_{\text{target}})\right)}. \tag{8}$$

This term promotes similarity in the mechanical capabilities of the two grasps.

**Jacobian-based Manipulability.** To capture the grasp's potential for object actuation, we execute the grasp in a differentiable simulator [56] and compute the Jacobian $J \in \mathbb{R}^{6 \times (n-6)}$ of the object's translational and rotational motion with respect to the hand's joint angles, excluding the translation and rotation of the hand base. Taking a max over the joint dimension yields a 6D vector summarizing the maximal motion in each direction $m = \max_j |J_j| \in \mathbb{R}^6$. The cost is then defined as the squared L2 distance between these maximal effect vectors:

$$d_{\text{jac}} = \|m_{\text{source}} - m_{\text{target}}\|^2. \tag{9}$$

This metric encourages the translated grasp to retain similar object-level controllability.

**Minibatch Approximation.** For each of the four cost functions above, we compute a separate entropic OT plan $\pi_\varepsilon^*$ using Sinkhorn's algorithm [81]. Following Tong et al. [84], we adopt a simulation-free approach to train the SB dynamics using these plans as static couplings. Since exact OT scales quadratically with sample size, we employ minibatch entropic OT [26, 28] to efficiently estimate the plans during training. Specifically, for each minibatch of source-target samples, we compute a local OT plan using the corresponding cost matrix and regularization parameter $\varepsilon$. These minibatch couplings serve as conditioning for generating Brownian bridge paths and estimating regression targets for score and flow matching. This strategy preserves the SB formulation's structure while enabling scalable learning in high-dimensional latent spaces.

### 4.3 Implementation Details

Our VAE is implemented using a PVCNN-based backbone [51]. For object shape representation, we encode the object point cloud using a LION VAE [89] pretrained on ShapeNet [13], which yields one global shape token and 512 local tokens. The hand observation latent is encoded into a single token. We further incorporate five contact anchor tokens, following the strategy introduced in [52], to guide translation via physical grounding. The latent Schrödinger Bridge is modeled using a U-ViT model [5], which serves as the backbone for $s_\theta$ and $v_\theta$.

For the Grasp Wrench Space Overlap cost, we reduce the 6D convex hull to its first three spatial dimensions, approximating the set of achievable center-of-mass forces. Monte Carlo estimation is used to evaluate the IoU of these 3D hulls efficiently on GPUs. For the Jacobian-based maximal effect descriptor, we leverage the Warp differentiable simulator [56] to simulate the grasp and compute the Jacobian of object pose with respect to joint angles. At inference time, latent samples are evolved using Euler-Maruyama integration of the learned SDE with discretized steps. No test-time finetuning is performed. Additional architectural and hyperparameter details are provided in the supplementary materials.

## 5 Experiments

### 5.1 Experimental Setup

**Dataset.** We evaluate our method on the MultiGripperGrasp dataset [12], a large-scale benchmark containing 30.4 million grasps across 11 robotic manipulators and 345 objects. For our experiments,

Table 1: Comparison of grasp translation methods on success rate, diversity, and 6D GWH IoU. Tasks include Human→Allegro, Human→Shadow, Shadow→Allegro, and the mean across them.

| Method | Success Rate (%)↑ | | | | Diversity (rad)↑ | | | | 6D GWH IoU (%)↑ | | | |
|---|---|---|---|---|---|---|---|---|---|---|---|---|
| | H→A | H→S | S→A | Mean | H→A | H→S | S→A | Mean | H→A | H→S | S→A | Mean |
| RFP [41] | 66.80 | 33.89 | 70.31 | 57.00 | 0.225 | 0.186 | 0.199 | 0.203 | 7.51 | 6.62 | 8.99 | 7.77 |
| Dex-Retargeting [72] | 56.93 | 16.21 | 56.44 | 43.19 | 0.221 | 0.186 | 0.196 | 0.201 | 5.67 | 4.41 | 6.97 | 5.68 |
| CrossDex [98] | 36.51 | 8.97 | 35.12 | 26.87 | 0.195 | 0.201 | 0.223 | 0.206 | 5.69 | 4.24 | 5.86 | 5.26 |
| Diffusion | 72.57 | 38.74 | 71.19 | 60.83 | **0.301** | 0.206 | 0.307 | 0.271 | 9.54 | 9.12 | 10.07 | 9.58 |
| Ours$_{pose}$ | 74.68 | 42.16 | 76.11 | 64.32 | 0.269 | 0.194 | 0.288 | 0.250 | 9.83 | 11.04 | 11.16 | 10.68 |
| Ours$_{contact}$ | 74.78 | 37.10 | 76.37 | 62.75 | 0.294 | **0.211** | **0.311** | **0.272** | **15.89** | **15.18** | 12.54 | 14.54 |
| Ours$_{GWH}$ | **77.73** | 42.59 | 78.74 | 66.34 | 0.293 | 0.197 | 0.309 | 0.266 | 15.09 | 14.14 | **14.67** | **14.63** |
| Ours$_{jacobian}$ | 77.23 | **45.15** | **79.98** | **67.45** | 0.278 | 0.205 | 0.292 | 0.258 | 11.16 | 9.77 | 11.73 | 10.88 |

we select 138 objects for training and 34 unseen objects for testing. Our primary evaluation scenarios focus on translating grasps between an articulated human hand (5 fingers, 20 DoFs), the Allegro hand (4 fingers, 16 DoFs), and the Shadow hand (5 fingers, 22 DoFs).

**Metrics.** We assess performance using several quantitative metrics. First, we report the grasp success rate using IsaacGym [57], following the GenDexGrasp [46] evaluation protocol. A grasp is considered successful if it maintains a stable hold on the object under six perturbation trials, where external forces are applied along $\pm x$, $\pm y$, and $\pm z$ axes. We apply each force as a uniform acceleration of $0.5m/s^2$ for 60 simulation steps and measure whether the object translates more than 2 cm. A grasp passes if it withstands all six trials. Simulation parameters include a friction coefficient of 10 and an object density of 10,000. We use IsaacGym's built-in positional controller to achieve the desired joint configurations. In addition to the success rate, we report the diversity of the generated grasps, computed as the average standard deviation across all revolute joints, as well as the functional similarity between source and translated grasps. The latter is measured using the full 6D IoU of GWH as defined in Eq. 8.

**Baselines.** We evaluate all four variants of our model, each trained using a distinct physics-based OT cost introduced in Sec. 4.2. As baselines, we compare against several methods. We include three recent grasp transfer approaches: RobotFingerPrint [41], which uses a unified gripper coordinate space; Dex-Retargeting [72], an optimization-based method that retargets key points on finger links; and the neural retargeting module from CrossDex [98]. Notably, both Dex-Retargeting and CrossDex require pre-defined, manually annotated link-to-link correspondences, which limit their generality for arbitrary hand morphologies. Finally, we compare against a generative baseline: a diffusion-based model trained with samples from Contact Map Similarity OT and conditioned on the source contact map.

## 5.2 Quality of Translated Grasps

We evaluate the overall quality of the translated grasps using two key metrics: success rate under physical perturbations and configuration diversity across generated samples. Table 1 (left and middle blocks) reports these metrics across three grasp translation tasks: Human→Allegro (H→A), Human→Shadow (H→S), and Shadow→Allegro (S→A), as well as their mean.

Our method significantly outperforms all baselines in terms of grasp success rate. Our approach using the Jacobian-based OT cost and GWH-based variants achieves the highest mean success rates (67.45% and 66.34%, respectively), which suggests that both wrench- and control-aligned transport costs are particularly effective for grasp transfer. The IK-style optimization baselines, Dex-Retargeting [72] and CrossDex Retargeting Module [98], perform poorly. Their focus on select links often ignores collisions with other parts of the hand (e.g., the palm), leading to a high rate of physically invalid grasps. In contrast, our generative approach learns to produce configurations that respect the full-hand morphology. Qualitative results in Fig. 4 (*left*) further illustrate that our method generates stable grasps even from sub-optimal source poses where baselines fail.

On the diversity metric, our contact-based variant achieves the highest mean diversity, producing a wide range of plausible grasps. This highlights that our methods can generate diverse grasp configurations while maintaining physical feasibility. Interestingly, other OT costs show lower diversity despite achieving high success rates, indicating that geometric alignment alone may lead to more conservative grasps. While the diffusion baseline is also highly diverse, our method achieves this while demonstrating superior performance on stability and functional alignment.

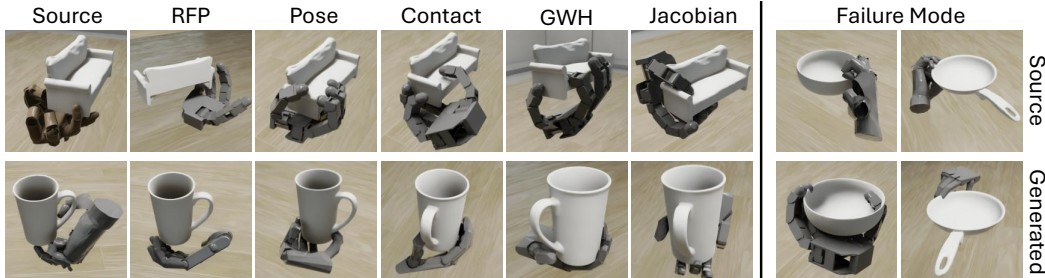

Figure 4: (*left*) **Qualitative results.** The 'Pose', 'Contact', 'GWH', and 'Jacobian' columns show results from our method when trained using the respective OT cost functions from Sec. 4.2. Our method generates stable and consistent grasps across different OT variants, even when the source grasp is sub-optimal, where RFP [41] fails. (*right*) **Failure Modes** where our method struggles with thin-shell objects due to challenging geometry.

In addition to performance, our approach offers significant computational advantages over optimization-based baselines. Methods like RobotFingerPrint [41] require costly per-grasp iterative optimization, leading to an average inference time of over 5 seconds. In contrast, our generative method amortizes this cost through batch processing. With 100 integration steps, our model achieves an average inference time of approximately 0.8 seconds per grasp, comparable to the diffusion baseline and substantially faster than RobotFingerPrint. While lightweight methods like CrossDex [98] are faster due to their single-pass architecture, our approach provides a strong balance between computational efficiency and the generation of high-quality, physically robust grasps. A more detailed efficiency report is included in Appendix D.2.

## 5.3 Physical Alignment between Translated Grasps

To evaluate whether our translated grasps preserve the functional intent of the source grasp, we compute the 6D GWH IoU between the source and the translated target grasps. As shown in Table 1 (right block), our methods trained with the GWH-based and Contact-based OT costs achieve the highest mean IoU by a substantial margin (14.63% and 14.54%, respectively), significantly outperforming all baselines. This indicates that our approach more effectively preserves the object-level physical capabilities of the grasp, rather than simply mimicking coarse finger geometry. We provide a complementary analysis on the geometric alignment of contact points in Appendix D.3, which further supports these findings.

Taken together, our results reveal a key trade-off between the different physics-informed objectives. The Jacobian-based cost yields the highest success rate, suggesting it is most effective at capturing the kinematic relationships required for stability. However, the GWH and Contact costs achieve superior functional alignment, as measured by GWH IoU. This indicates that directly optimizing for force-exertion capabilities (GWH) or contact patterns (Contact) is more effective for transferring the grasp's underlying mechanical intent, even if this sometimes results in a slightly lower stability score compared to a pure manipulability-based objective.

## 5.4 Ablation Study

We conduct an ablation study to assess the sensitivity of our method to two key hyperparameters in the Schrödinger Bridge formulation: the magnitude of the diffusion rate $\sigma$, and the number of discretization steps used during Euler–Maruyama integration at test time. Both ablations are performed on the H→A task using the Contact Map OT model, and results are reported in Tables 2 and 3.

**Effect of Diffusion Rate.** We vary the diffusion rate $\sigma \in \{0.01, 0.1, 1.0\}$ and observe that smaller diffusion magnitudes lead to better grasp success and GWH IoU. Larger diffusion ($\sigma = 1.0$) produces noisier paths and degrades both physical precision and grasp quality, despite yielding higher diversity. This supports the intuition that overly noisy reference processes can disrupt fine-grained transport and hinder alignment with physical intent.

**Effect of Discretization Steps.** We evaluate our method under varying numbers of time steps during test-time integration: $\{10, 100, 200\}$. As shown in Table 3, using 100 steps offers a strong tradeoff between accuracy and efficiency, while more steps lead to better physical alignment. Interestingly, our method remains robust even with as few as 10 steps, outperforming the diffusion baseline across

Table 2: Effect of diffusion rate $\sigma$ on performance (H→A).

| $\sigma$ | Success %↑ | Div. (rad)↑ | IoU↑ |
|---|---|---|---|
| 0.01 | **77.16** | 0.284 | **16.72** |
| 0.1 | 74.78 | 0.294 | 15.89 |
| 1.0 | 64.39 | **0.378** | 13.83 |

Table 3: Effect of number of discretization steps on performance (H→A) from our method and the diffusion baseline.

| # Steps | Success %↑ | | Div. (rad)↑ | | IoU↑ | |
|---|---|---|---|---|---|---|
| | Ours | Diff. | Ours | Diff. | Ours | Diff. |
| 10 | 71.45 | 60.00 | 0.278 | **0.344** | 13.80 | 6.19 |
| 100 | **75.12** | 72.57 | 0.283 | 0.301 | 14.00 | 9.54 |
| 200 | 74.78 | 73.17 | 0.294 | 0.297 | **15.89** | 9.66 |

all metrics and steps. In contrast, the diffusion model's performance degrades substantially under low discretization, highlighting the benefit of our SB-based formulation, which yields more stable and sample-efficient generation dynamics.

## 6 Discussion

In this work, we present a novel formulation of vision-based dexterous grasp translation as a Schrödinger Bridge problem between grasp distributions. By leveraging a latent score and flow matching framework and designing physics-informed ground costs, our method enables simulation-free, distribution-level grasp transfer across robotic hands with distinct morphologies. Empirical evaluations across several hand-object settings demonstrate that our approach not only produces stable and diverse grasps but also preserves functional grasp properties more effectively than object-agnostic or geometry-based baselines. These results highlight the promise of distributional transport techniques for advancing semantic grasp understanding and generalization in robotic manipulation.

**Limitations.** Despite its strengths, our approach has some limitations. Most notably, as shown in Fig. 4 (*right*), the performance drops in scenarios involving thin-shell objects with ambiguous or minimal contact regions, where the valid grasp configuration space is largely confined. Additionally, we observe that the results for the Shadow hand are consistently lower than for the Allegro hand across multiple metrics, which aligns with prior findings in the MultiGripperGrasp dataset [12] due to its more complex kinematics and larger workspace. Furthermore, our current framework does not generalize to unseen hand morphologies at inference time. The VAE decoder implicitly learns the kinematic structure of the target hand, and would thus require retraining or fine-tuning on data from a new hand to generate valid grasps. Finally, these discrepancies suggest that improving dataset quality and coverage for specific hands may further enhance generalization and translation fidelity.

**Societal Impact.** Finally, while our framework is designed for general-purpose manipulation, it introduces broader implications for deployment in real-world systems. Effective grasp translation could accelerate learning from heterogeneous demonstrations or expand manipulation capabilities in assistive robotics. However, it also raises considerations around safe deployment, failure detection, and responsible generalization across users and hardware. We encourage future work to examine how distributional grasp priors can be integrated into transparent and human-aligned manipulation pipelines.

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

# A  Background on Schrödinger Bridges and Simulation-Free Training

## A.1  The Schrödinger Bridge Problem

The Schrödinger Bridge (SB) problem seeks a stochastic process $\mathbb{P}$ that evolves from a source distribution $q_0(x)$ to a target distribution $q_1(x)$ while being as "close" as possible to a reference process $\mathbb{Q}$ (typically Brownian motion). This is formally expressed as minimizing the Kullback-Leibler (KL) divergence between the path measures:

$$\mathbb{P}^* = \underset{\mathbb{P}:p_0=q_0,p_1=q_1}{\arg\min} \ \mathrm{KL}(\mathbb{P}||\mathbb{Q}). \tag{10}$$

The optimal process $\mathbb{P}^*$ can be characterized as a mixture of Brownian bridges weighted by an entropically regularized optimal transport (OT) plan [11, 21]. As shown in Eq. 1 in the main text, this plan $\pi_\varepsilon^*$ finds a low-cost coupling between $q_0$ and $q_1$. The work of Föllmer [31] shows that if $\mathbb{Q}$ is Brownian motion, the SB is uniquely described by a mixture of Brownian bridges drawn according to $\pi_\varepsilon^*$. This insight is crucial, as it reduces a complex dynamic optimization problem to a more tractable computation over static distributions.

## A.2  Learning Stochastic Dynamics with [SF]$^2$M

The dynamics of the SB are modeled by the Itô SDE in Eq. 2. The marginal densities $p_t(x)$ induced by this SDE evolve according to the Fokker-Planck equation [73]:

$$\frac{\partial p_t}{\partial t} = -\nabla \cdot (p_t u_t) + \frac{g(t)^2}{2} \Delta p_t, \tag{11}$$

where $\Delta p_t = \nabla \cdot (\nabla p_t)$ is the Laplacian. The probability flow ODE in Eq. 3 shares the same marginals $p_t(x)$. To learn these dynamics from data without direct access to the marginals $p_t$, the [SF]$^2$M framework [84] leverages the connection to entropic OT. By sampling a source-target pair $(x_0, x_1)$ from the entropic OT plan $\pi_\varepsilon^*$, one can construct a conditional Brownian bridge path between them. For a constant diffusion rate $g(t) = \sigma$, the conditional distribution $p_t(x \mid x_0, x_1)$ is a Gaussian $\mathcal{N}(x; (1-t)x_0 + tx_1, \sigma^2 t(1-t)I)$. From this, we obtain closed-form expressions for the conditional score and probability flow drift, which serve as regression targets for the neural networks:

$$u_t^\circ(x|x_0, x_1) = \frac{1-2t}{t(1-t)}(x - ((1-t)x_0 + tx_1)) + (x_1 - x_0),$$
$$\nabla \log p_t(x|x_0, x_1) = \frac{(1-t)x_0 + tx_1 - x}{\sigma^2 t(1-t)}. \tag{12}$$

These targets are used in the loss function shown in Eq. 4.

## A.3  Weighting Schedule $\lambda(t)$

Following Tong et al. [84], we apply a time-dependent weighting schedule $\lambda(t)$ to stabilize the score regression loss across time. This is essential because the conditional score $\nabla_x \log p_t(x \mid x_0, x_1)$ grows unbounded as $t \to 0$ or $t \to 1$, leading to an imbalance in the loss. To address this, we adopt the formulation that standardizes the regression target to have unit variance. This is done by predicting the noise added in sampling $x_t$ from the linear interpolation $\mu_t = (1-t)x_0 + tx_1$ under a Brownian bridge with variance $\sigma_t^2 = \sigma^2 t(1-t)$. This leads to:

$$\lambda(t) = \sigma_t = \sigma\sqrt{t(1-t)}.$$

This setting ensures that the target score $\nabla_x \log p_t(x \mid x_0, x_1)$ corresponds to $\epsilon/\sigma_t$, with $\epsilon \sim \mathcal{N}(0, I)$, making the scaled target distributed as standard Gaussian.

In practice, we use an alternative numerically stable formulation derived by rescaling the target by $\frac{1}{2}\sigma_t^2 \nabla_x p_t$, which yields the weighting schedule:

$$\lambda(t) = \frac{2}{\sigma^2}\sigma_t = \frac{2\sqrt{t(1-t)}}{\sigma}. \tag{13}$$

Under this form, the squared score loss term simplifies to:

$$\lambda(t)^2 \left\| s_\theta(t, x) - \nabla \log p_t(x|x_0, x_1) \right\|^2 = \left\| \lambda(t)s_\theta(t, x) + \epsilon_t \right\|.$$

This avoids any division by small values resulting from boundary conditions and improves numerical stability. All models in our experiments were trained using this second formulation and the simplified regression objective:

$$\mathcal{L}(\theta) = \mathop{\mathbb{E}}_{\substack{t \sim \mathcal{U}(0,1) \\ x \sim p_t(x|x_0,x_1) \\ (x_0,x_1) \sim \pi^*}} \left[ \|v_\theta(t,x) - u_t^\circ(x|x_0,x_1)\|^2 + \|\lambda(t)s_\theta(t,x) + \epsilon_t\|^2 \right]. \tag{14}$$

## B  Architecture and Training Details

### B.1  VAE Training

Our VAE consists of two main components: a point cloud encoder for the segmented hand point cloud, and an MLP-based latent autoencoder that encodes and decodes grasp configurations.

**Hand Encoder.** We use a PVCNN-based architecture [51] to process the input hand point cloud. The input to the encoder is a point cloud of size $N \times 3$. The PVCNN follows a hierarchical structure with point and voxel convolution blocks as described below:

Table 4: PVCNN Point Feature Blocks

| Output Channels | Num Layers | Kernel Size | Voxel Resolution Multiplier |
|:---:|:---:|:---:|:---:|
| 64 | 1 | 32 | ✓ |
| 64 | 2 | 16 | ✓ |
| 128 | 1 | 16 | ✓ |
| 1024 | 1 | N/A | ✗ |

The final 1024-dimensional global features are passed through an MLP consisting of [256, 128] hidden units, followed by max pooling to produce a single global feature vector for each input.

**Grasp Configuration VAE.** The latent grasp representation is modeled using a fully connected variational autoencoder. The encoder receives the hand representation and maps it into a latent space of size 768. The decoder reconstructs the full hand joint configuration. Layer sizes are as follows:

Table 5: Latent VAE Layer Sizes

| Network | Input | Hidden Layers | Latent Dim | Output Dim |
|:---:|:---:|:---:|:---:|:---:|
| Encoder | 1472 | [2048, 1024] | 768 | – |
| Decoder | 768 | [2048, 1024, 256] | – | 9 + # DoFs |

**Training Configuration.** We train the VAE for 18 epochs using a batch size of 256, Adam optimizer with a learning rate of $3 \times 10^{-4}$, and 16 dataloader workers. The loss function includes a weighted combination of reconstruction losses with $\alpha = 0.6$ and a KL divergence term with a very small weight ($\beta = 1 \times 10^{-5}$). The training time is approximately one GPU day on a single NVIDIA A6000 GPU.

### B.2  Score and Flow Model Architecture

Both the score network $s_\theta$ and the flow network $v_\theta$ in our pipeline share the same U-ViT-based backbone architecture [5]. The input to each model consists of a concatenation of latent hand representations, object shape encodings, and contact anchor tokens, following the conditioning design described in Sec. 4.3 of the main paper.

**U-ViT Backbone.** We use a vision transformer architecture adapted for point-based data. The network is composed of 12 transformer blocks, each with multi-head self-attention and MLP layers. Time conditioning is injected via learned embeddings scaled by a temperature factor. The configuration is detailed in Table 6.

Table 6: U-ViT Backbone Configuration

| Parameter | Value |
|---|---|
| Number of Transformer Layers (Depth) | 12 |
| Embedding Dimension | 512 |
| Number of Attention Heads | 8 |
| MLP Expansion Ratio | 4 |
| Time Embedding Scale | 1000.0 |
| Dropout Rate | 0.0 |
| Attention Dropout Rate | 0.1 |

**Input Dimensions.** The model is conditioned on several input tokens:

- **Latent Hand Token:** 768-dimensional vector produced by the VAE encoder.
- **Global Object Token:** 128-dimensional latent from the pretrained LION object encoder.
- **Local Object Tokens:** 2048 points, each with 4 dimensions (XYZ + latent code).
- **Contact Anchor Tokens:** 5 tokens computed by applying farthest point sampling to object points identified via thresholding (within 0.005 m) to the nearest point to the hand point cloud.

**Training Configuration.** We train both the score and flow networks using the loss in Eq. 14 with the weighting schedule described in Appendix A.3. The training setup is summarized in Table 7. The training time is approximately 650 GPU hours on NVIDIA L40 GPUs.

Table 7: Training Hyperparameters for Score and Flow Models

| Hyperparameter | Value |
|---|---|
| Total Steps | 20000 |
| Batch Size (on each GPU) | 512 |
| Learning Rate | $2 \times 10^{-4}$ |
| Linear Warmup Steps | 256 |
| Gradient Clipping | 1.0 |
| EMA Decay | 0.999 |
| EMA Start Step | 10000 |
| $\sigma$ (used in Table. 1) | 0.1 |

## C  OT Cost Formulation and Computation

We implement four distinct ground costs for the optimal transport (OT) coupling in our framework. Two of these (Euclidean base pose and contact map Chamfer distance) are computed in closed form using standard metrics in $\mathbb{R}^n$. The remaining two involve more complex geometry- and physics-informed reasoning.

**Grasp Wrench Space Overlap.** The grasp wrench space (GWS) [29] of a grasp represents the set of all wrenches (forces and torques in $\mathbb{R}^6$) that the grasp can apply to an object through contact. For a grasp with $k$ contact points $\{c_i\}_{i=1}^k$ and associated contact normals $\{n_i\}$, we construct a set of unit wrench vectors $\{w_i\}$ as follows:

$$w_i = \begin{bmatrix} f_i \\ c_i \times f_i \end{bmatrix} \in \mathbb{R}^6, \quad f_i = \alpha_i n_i, \ \alpha_i > 0.$$

We form the convex hull of these wrenches to approximate the GWS, denoted as the grasp wrench hull (GWH). The distance between two grasps is then defined as 1 minus the intersection-over-union (IoU) of their GWH volumes:

$$d_{\text{wrench}} = 1 - \frac{\text{Vol}(\text{Hull}(GWS_{\text{source}}) \cap \text{Hull}(GWS_{\text{target}}))}{\text{Vol}(\text{Hull}(GWS_{\text{source}}) \cup \text{Hull}(GWS_{\text{target}}))}.$$

Since computing the exact intersection and union volumes of 6D convex hulls is intractable in closed form, we approximate the GWH IoU using GPU-accelerated Monte Carlo sampling. We sample a large number of points (typically $O(10^6)$) in the 6D bounding box enclosing both hulls, and compute membership in each hull using their supporting hyperplane inequalities. Intersection and union volumes are then estimated as:

$$\text{Vol}_{\text{intersect}} \approx \frac{\text{\#pts in both}}{\text{\#samples}} \text{Vol}_{\text{box}}.$$

A 3D implementation of this approximation is used to compute batch-wise GWH IoU efficiently during training, and the full 6D implementation is reported for the experiments.

**Jacobian-based Manipulability.** To capture the potential of a grasp to actuate the object, we compute the Jacobian matrix $J \in \mathbb{R}^{6 \times (n-6)}$ relating hand joint velocities to object motion. The Jacobian is computed using the Warp differentiable simulator [56] following the setup from Turpin et al. [87] with fixed object and grasp configurations. We exclude the 6-DoF base pose of the hand, focusing only on internal joints.

Each column $J_j$ of the Jacobian represents the effect of joint $j$ on the object's linear and angular velocity. We summarize the object-level actuation capability of a grasp by computing a 6D vector $m$ of maximal manipulability across joints:

$$m = \max_j |J_j| \in \mathbb{R}^6.$$

The OT cost is then defined as the squared $\ell_2$ distance between source and target maximal actuation vectors:

$$d_{\text{jac}} = \|m_{\text{source}} - m_{\text{target}}\|^2.$$

This cost encourages the translated grasp to maintain similar controllability over object motion.

# D  Additional Results

## D.1  Qualitative Examples Across Tasks

Figure 5 presents additional qualitative examples across all three grasp translation tasks: Human→Allegro, Human→Shadow, and Shadow→Allegro. Our method remains robust and produces stable, contact-consistent grasps even when the source grasp is suboptimal or incomplete. In contrast, the object-agnostic baseline RFP often fails to recover valid contact configurations, particularly for complex hand-object interactions or geometrically mismatched hands.

## D.2  Inference Time Comparison

We compare the amortized inference time of our method against all baselines in Table 8. While our framework is designed primarily as an offline planner, its inference speed is competitive.

As shown in the table, our method takes approximately 0.8 seconds to generate a grasp using 100 Euler-Maruyama steps. This is comparable to the diffusion baseline (0.5s). The iterative sampling process, where each of the 100 steps requires a full forward pass through our U-ViT model, is the primary factor for this timing. In contrast, lightweight methods like CrossDex are significantly faster, as they only require a single forward pass through a small MLP. Nevertheless, our approach is substantially faster than the optimization-based RobotFingerPrint, which requires over 5 seconds per grasp due to its costly per-instance optimization procedure.

Table 8: Average inference time per grasp (amortized, in seconds). Our method is comparable to the diffusion baseline and significantly faster than optimization-based approaches.

| Method | Time (sec/grasp) | | | |
| --- | --- | --- | --- | --- |
| | H→A | H→S | S→A | Mean |
| CrossDex Retargeting [98] | 1.1e-5 | 1.1e-5 | 1.1e-5 | 1.1e-5 |
| Dex-Retargeting [72] | 0.13 | 0.16 | 0.08 | 0.12 |
| Diffusion (100 steps) | 0.5 | 0.5 | 0.5 | 0.5 |
| Ours (100 steps) | 0.8 | 0.8 | 0.8 | 0.8 |
| RobotFingerPrint [41] | 5.2 | 5.8 | 5.1 | 5.37 |

### D.3 Contact Alignment

In addition to the GWH IoU metric presented in the main text, we evaluate the geometric alignment of contact patterns using a Contact Alignment metric. This metric is defined as the squared Chamfer distance between the source and target contact maps on the object surface, where a lower value indicates better alignment. This serves as a useful proxy for how well the local interaction geometry of the grasp is preserved during translation.

Table 9 presents the results. Our model trained with the Contact OT cost ($Ours_{contact}$) achieves a mean distance of just 4.99 cm$^2$, significantly outperforming all other methods. This quantitatively confirms that this variant is highly effective at preserving the precise locations of contact, which is a critical precondition for achieving a functionally similar and physically stable grasp. The other baselines struggle to align contact points, resulting in much larger distances and, as shown in the main paper, lower success rates.

Table 9: Comparison on Contact Alignment (cm$^2$)↓. Lower values indicate better alignment between the source and target grasp's contact points on the object surface.

| Method | Contact Alignment (cm$^2$)↓ | | | |
| --- | --- | --- | --- | --- |
| | H→A | H→S | S→A | Mean |
| RobotFingerPrint [41] | 19.21 | 16.66 | 11.27 | 15.71 |
| Dex-Retargeting [72] | 9.53 | 12.92 | 6.68 | 9.71 |
| CrossDex Retargeting [98] | 9.85 | 14.52 | 7.06 | 10.48 |
| Ours (Contact OT) | **5.05** | **3.37** | **6.54** | **4.99** |
| Ours (GWH OT) | 10.71 | 7.26 | 10.19 | 9.39 |

### D.4 VAE Ablation Study

To validate the design choices for our Variational Autoencoder (VAE), we conduct an ablation study comparing our full model against three alternatives:

- **PointNet VAE**: A baseline inspired by [1] that reconstructs the hand point cloud via Chamfer distance and decodes hand parameters in a separate branch.
- **Ours w/o Mesh Loss**: Our VAE trained only on the hand parameter reconstruction loss ($\mathcal{L}_{VAE} = \mathbb{E}[\|\hat{g} - g\|^2]$), without the differentiable FK layer and mesh vertex loss.
- **Ours w/o Param Loss**: Our VAE trained only on the mesh vertex loss ($\mathcal{L}_{VAE} = \mathbb{E}[\|\hat{v} - \text{FK}(g)\|^2]$), without the direct parameter reconstruction loss.

We evaluate reconstruction quality using the L2 error on the decoded hand parameters, tested on a held-out set of objects. As shown in Table 10, our proposed VAE architecture significantly outperforms all alternatives.

Table 10: Evaluation of VAE architectures using Hand Parameters L2 Error↓.

| Method | Allegro | Shadow | Human |
| --- | --- | --- | --- |
| PointNet VAE | 0.1965 | 0.2001 | 0.2468 |
| Ours w/o Mesh Loss | 0.0967 | 0.0933 | 0.1270 |
| Ours w/o Param Loss | 0.0439 | 0.0441 | 0.0534 |
| Ours (Full) | **0.0335** | **0.0286** | **0.0523** |

Our analysis indicates that the PointNet VAE struggles because the Chamfer distance loss can be indiscriminative; it may achieve a low value even if the reconstructed pose is incorrect but has large overlapping regions with the ground truth. Ablating the mesh loss harms the learning of fine-grained geometric details, while ablating the parameter loss can lead to memorization issues, as the high-capacity latent space (dim=768) can overfit to the relatively low-dimensional hand configuration space (typically ∼20 DoFs).

### D.5  Results on Additional Transfer Settings

To provide a more comprehensive evaluation, we run experiments on the three remaining transfer settings: Allegro→Human (A→H), Shadow→Human (S→H), and Allegro→Shadow (A→S).

We note a technical limitation in reporting the IsaacGym success rate for tasks where the human hand is the target. Our human hand is an articulated model based on MANO [74] with non-watertight link meshes, which causes instabilities in the IsaacGym simulator. We are therefore unable to report success rates for the A→H and S→H settings.

The results for success rate (A→S only), diversity, and 6D GWH IoU are presented in Tables 11, 12, and 13, respectively. Our method consistently outperforms the baselines in preserving functional intent, achieving significantly higher GWH IoU, particularly when transferring to the complex human hand. We also generate a more diverse set of plausible grasps, highlighting the flexibility of our learned transport plan. Finally, in the A→S setting where a direct stability comparison was possible, our method achieved the highest success rate, demonstrating superior physical plausibility. These findings show that the advantages of our approach are robust across these challenging transfer settings.

Table 11: Comparison on Success Rate (%)↑ for A→S.

| Method | A→S |
|---|---|
| RobotFingerPrint | 30.27 |
| Diffusion | 33.96 |
| Ours (Contact OT) | 38.00 |
| Ours (GWH OT) | **41.49** |

Table 12: Comparison on Diversity (rad)↑.

| Method | Diversity (rad)↑ | | | |
|---|---|---|---|---|
| | A→H | S→H | A→S | Mean |
| RobotFingerPrint | 0.196 | 0.180 | 0.182 | 0.186 |
| Diffusion | 0.283 | 0.298 | 0.207 | 0.263 |
| Ours (Contact OT) | **0.317** | **0.332** | **0.212** | **0.287** |
| Ours (GWH OT) | 0.310 | 0.327 | 0.178 | 0.272 |

Table 13: Comparison on 6D GWH IoU (%)↑.

| Method | 6D GWH IoU (%)↑ | | | |
|---|---|---|---|---|
| | A→H | S→H | A→S | Mean |
| RobotFingerPrint | 3.46 | 4.11 | 3.51 | 3.69 |
| Diffusion | 10.41 | 11.82 | 8.78 | 10.34 |
| Ours (Contact OT) | **13.59** | **15.37** | 8.87 | **12.61** |
| Ours (GWH OT) | 11.76 | 13.30 | **9.16** | 11.41 |

## E  Compute Resources

Our experiments were conducted on a combination of local servers and a high-performance computing cluster. The local server consists of a 24-core CPU and 2 NVIDIA A6000 GPUs, which were used for model development, ablation studies, and VAE training. For large-scale training, we utilized a compute cluster, where each cluster node is equipped with two 26-core CPUs and 8 NVIDIA L40 GPUs.

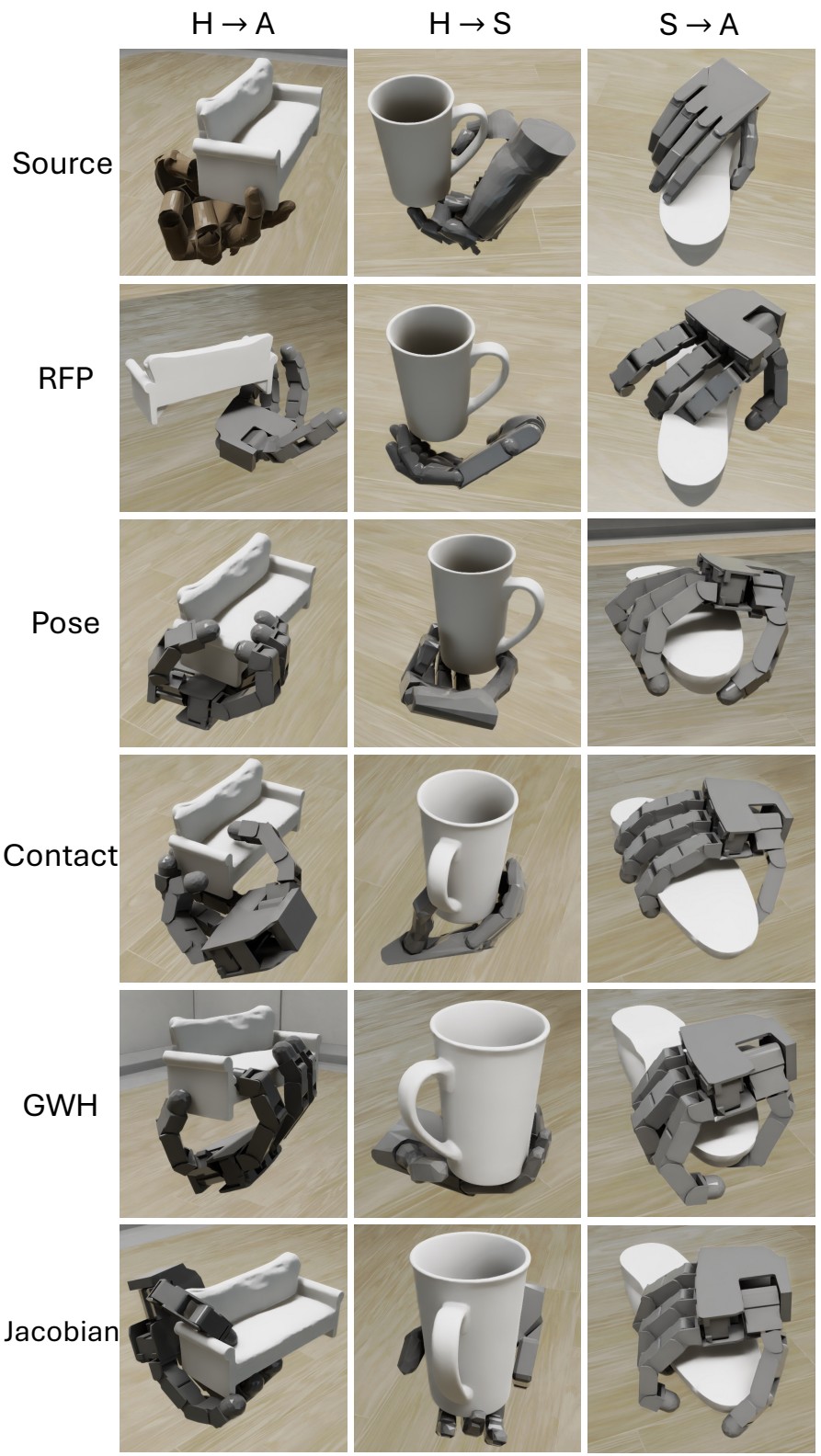

Figure 5: **More qualitative examples.** Each column shows a source grasp and its translated output for different methods. Our method consistently recovers physically plausible grasps across tasks, while RFP frequently fails, especially with noisy source grasps.

