# OpenReview forum: "Grasp2Grasp: Vision-Based Dexterous Grasp Translation via Schrödinger Bridges"
_NeurIPS.cc/2025/Conference — NeurIPS 2025 poster_

### Official Review · Reviewer_4u79 · 2025-06-18

**Clarity:** 3
**Significance:** 4
**Originality:** 3
**Rating:** 5
**Confidence:** 3

**Summary:**

This paper addresses the problem of transferring successful grasps from one robot hand to another based on the well-known Simulation-Free Schrödinger Bridges. The method first encodes grasps from each hand into a shared latent space, then uses the Schrödinger Bridge to model the optimal transport between the two latent distributions. Extensive experiments show that the proposed approach significantly outperforms existing grasp transfer methods.

**Questions:**

- Would it be possible to include the remaining three transfer settings (A→H, S→H, and A→S)? Including these would make the evaluation more completed and reduce potential bias.

**Ethical Concerns:**

["NO or VERY MINOR ethics concerns only"]

**Final Justification:**

My concerns have been addressed. I encourage the authors to consider and address all reviewers' feedback in the final version. I maintain my positive rating.

**Limitations:**

Yes.

**Paper Formatting Concerns:**

No.

**Quality:**

4

**Strengths And Weaknesses:**

**Strengths:**
- Overall, this is a well-written and insightful paper that addresses the long-standing challenge of transferring dexterous grasp.

- The use of Simulation-Free Schrödinger Bridges for transferring grasp is both novel and well-motivated.

- Experiments on the MultiGripperGrasp dataset demonstrate substantial improvements over state-of-the-art methods.

- The ablation studies (Tables 1–3) further validate the effectiveness of the proposed optimal transport cost design.

**Weaknesses:**

I have no major concerns about the paper. However, here are some suggestions to improve clarity and completeness:
-  *Lack of evaluation of the VAE*. Although the VAE is used primarily as a means to encode grasps across different robot hands, its performance should still be evaluated, for instance, reconstruction quality. Additionally, some visualization of the latent space trajectories during the Simulation-Free Schrödinger Bridge process would help further understand the proposed solution.
- *Concerns over presentation.* While the paper is generally well-written, I find Sections 3.1 and 3.2 to be somewhat redundant, as they closely follow the original Simulation-Free Schrödinger Bridges paper [78]. These sections could be reduced or moved to the appendix. In contrast, the core technical contribution—modeling the Schrödinger Bridge between latent grasp spaces via $v_\theta(t, z)$ and $s_\theta(t, z)$—is essential to the paper but is currently under-explained. I recommend moving this part to the main text to better clarity.

---

> ### Author Rebuttal · Authors · 2025-07-31
>
> We sincerely thank the reviewer for their time and insightful feedback. We are encouraged that the reviewer found our manuscript "insightful" and our method "novel and well-motivated." We address each point below.
>
> > *W1*: On VAE Evaluation and Latent Space Visualization.
>
> Thank you for these concrete suggestions to improve the paper's completeness.
>
> **VAE Evaluation**
>
> We agree that an evaluation of the VAE is important. We have conducted an additional ablation study to validate our design choices. We compare our full VAE against three alternatives:
> - A **PointNet VAE** baseline based on [1] that reconstructs the hand point cloud via Chamfer distance and decodes hand parameters in a separate branch.
> - **Ours w/o Mesh Loss**: Our VAE trained only on the hand parameter reconstruction loss, without the FK layer and mesh vertex loss.
> - **Ours w/o Param Loss**: Our VAE trained only on the mesh vertex loss, without the direct parameter loss.
>
> We evaluate performance using the Hand Parameters L2 Error on a held-out set of objects. We choose this metric for evaluating reconstruction quality because our VAE ultimately decodes the hand parameters. As shown in the table below, our proposed VAE architecture significantly outperforms all alternatives.
>
> **Table 1: Evaluation for VAE: Hand Parameters L2 Error**
> |                     | Allegro | Shadow | Human  |
> |---------------------|---------|--------|--------|
> | PointNet VAE        | 0.1965  | 0.2001 | 0.2468 |
> | Ours w/o Mesh Loss  | 0.0967  | 0.0933 | 0.1270 |
> | Ours w/o Param Loss | 0.0439  | 0.0441 | 0.0534 |
> | Ours                | **0.0335**  | **0.0286** | **0.0523** |
>
> Our analysis indicates that the PointNet VAE struggles because the Chamfer distance loss is often indiscriminative, as evidenced by many works [1]. It can achieve a low value even when the reconstructed grasp pose is incorrect but has large overlapping regions. Ablating the mesh loss harms performance on fine-grained details, while ablating the parameter loss can lead to memorization issues given the high capacity of the latent space, as the size of the hand configuration space (typically ~20) is relatively small compared to the size of the latent space (e.g., 768 in our case). We will include this table and analysis in the appendix of the revised manuscript.
>
> **Latent Trajectory Visualization**
>
> This is an excellent idea. Direct visualization of the latent trajectory by decoding intermediate points (i.e., for $t$ between 0 and 1) is unfortunately not straightforward. While the latent space is shared, the VAE decoder is hand-specific due to the use of the forward kinematics (FK) layer. Decoding an intermediate latent $z_t$ would be ill-defined, as it is unclear which hand's FK layer should be used for visualization.
>
> However, to provide the intuition the reviewer requested, we will add a new figure to Section 4 showing the learned transport process on a 2D toy example (e.g., translating from a two-moons distribution to an 8-Gaussians distribution). This will visually illustrate the latent evolution concept that underpins our method.
>
> > *W2*: Presentation of core technical contribution.
>
> We appreciate this valuable feedback on improving the paper's structure and clarity. We will implement the following changes:
> 1. We will streamline Sections 3.1 and 3.2, moving the general background on Schrödinger Bridges (SB) and the derivation of the SF$^2$M to the appendix. We will retain only the key equations and concepts in the main text that are directly referenced in our methodology (e.g., the entropic OT plan and the final conditional loss function).
> 2. We will use the freed-up space to expand Section 4.1, providing a more detailed explanation of our core technical contribution: modeling the SB between latent grasp spaces. This new text will elaborate on how the neural networks $v_{\theta}(t,z)$ and $s_{\theta}(t,z)$ approximate the conditional flow and score, how they are trained using paired samples drawn from the OT plan, and how they are used at inference time to evolve a source latent code to a target one.
> 3. The new 2D visualization figure mentioned above will be included in this revised section to make the explanation more intuitive.
>
> > *Q1*: Remaining three transfer settings (A→H, S→H, and A→S)
>
> Yes, this is a great suggestion to make our evaluation more comprehensive. We have run experiments on the remaining three transfer settings: Allegro→Human (A→H), Shadow→Human (S→H), and Allegro→Shadow (A→S).
>
> We must note a technical limitation: we were unable to report the *IsaacGym success rate* for settings where the target is the human hand (A→H and S→H). This is because our human hand model is an articulated version of the MANO model [2] with non-watertight link meshes, which causes instability in the IsaacGym simulator. Porting our evaluation to a simulator capable of handling such assets (e.g., IsaacSim) is beyond the current scope of this work and is planned for future development.
>
> However, we are able to report the success rate for A→S and the other metrics (Diversity, 6D GWH IoU) for all three new settings. The results are presented below.
>
> ***Table 2: Comparison on Success Rate (%)↑***
> |     | RobotFingerPrint | Diffusion | Ours (Contact OT) | Ours (GWH OT) |
> |-----|------------------|-----------|-------------------|---------------|
> | A→S | 30.27            | 33.96     | ***38.00***             | **41.49**         |
>
> ***Table 3: Comparison on Diversity (rad)↑***
> |                   | A→H   | S→H   | A→S   | Mean  |
> |-------------------|-------|-------|-------|-------|
> | RobotFingerPrint  | 0.196 | 0.180 | 0.182 | 0.186 |
> | Diffusion         | 0.283 | 0.298 | ***0.207*** | 0.263 |
> | Ours (Contact OT) | **0.317** | **0.332** | **0.212** | **0.287** |
> | Ours (GWH OT)     | ***0.310*** | ***0.327*** | 0.178 | ***0.272*** |
>
> ***Table 4: Comparison on 6D GWH IoU (%)↑***
> |                   | A→H  | S→H  | A→S  | Mean |
> |-------------------|------|------|------|------|
> | RobotFingerPrint  | 3.46 | 4.11 | 3.51 | 3.69 |
> | Diffusion         | 10.41 | 11.82 | 8.78 | 10.34 |
> | Ours (Contact OT) | **13.59** | **15.37** | ***8.87*** | **12.61** |
> | Ours (GWH OT)     | ***11.76*** | ***13.30*** | **9.16** | ***11.41*** |
>
> Our method consistently outperforms both baselines in preserving functional intent, achieving significantly higher 6D GWH IoU (Table 4), particularly when transferring to the more complex human hand. We also generate a more diverse set of plausible grasps (Table 3), highlighting the flexibility of our learned transport plan. Finally, in the A→S setting where a direct stability comparison was possible, our method achieved the highest success rate (Table 2), demonstrating superior physical plausibility. These findings show that the advantages of our approach are robust, even in these challenging transfer settings.
>
> We will add these tables, along with the analysis and corresponding qualitative results, to the appendix to provide a more complete picture of our method's performance.
>
> #### Reference:
> [1] Achlioptas et al., "Learning representations and generative models for 3d point clouds." ICML 2018.
>
> [2] Romero et al., "Embodied hands: Modeling and capturing hands and bodies together." SIGGRAPH ASIA 2017.

---

> ### Comment · Reviewer_4u79 · 2025-08-05
>
> Thank you for your clear and detailed rebuttal. With the revised manuscript incorporating more comprehensive evaluations—such as latent visualizations, additional baselines, extended experimental setups, and a discussion on inference speed—I believe the paper is of high quality. I have no further questions.

---

### Official Review · Reviewer_Xi3Y · 2025-06-30

**Clarity:** 2
**Significance:** 2
**Originality:** 3
**Rating:** 3
**Confidence:** 3

**Summary:**

The paper introduces Grasp2Grasp, a method for grasp pose translation that aims to transfer grasping poses from one embodiment to another. Grasp2Grasp adopts a novel strategy by modeling valid grasps as an object-conditioned distribution and learning a distributional transformation between different embodiments. The translation process involves sampling from the latent space of the source embodiment, transforming the latent representation to align with the target embodiment's distribution, and decoding it into a corresponding grasp pose. To ensure the plausibility of the translated grasps, the method incorporates several cost terms that measure differences between grasp poses not only in appearance but also in terms of task effectiveness. Experimental results demonstrate the method’s ability to generate plausible and task-relevant grasping poses across embodiments.

**Questions:**

- Does the encoding process take the grasping pose into account? According to the formulation $z_0\sim q_0(z_{source}\vert o_{obj})$, only the object observation is used. However, Figure 2 suggests that the observation also includes the hand pose.
- The roles of the source hand pose gsg_sgs in Algorithms 2 and 3 are unclear. It appears that $g_s$ is not involved in either the encoding or the sampling process.
- What metric is used to compute the distance between the two 6D rotation representations in Equation 9?
- Are the generated grasping poses validated in a physics simulator?

**Ethical Concerns:**

["NO or VERY MINOR ethics concerns only"]

**Final Justification:**

Thank you to the authors for their efforts during the rebuttal period. Unfortunately, my main concerns about the qualitative results remain unresolved. Based on the original results—particularly Figure 1 and Figure 4 in the supplementary material—I am not convinced that the transferred grasps preserve functional similarity to the original grasps or that they are physically plausible. In addition, the diversity of object geometries is limited.

The rebuttal presents additional quantitative results and comparisons. However, I would regard the contribution as a substantial improvement only if both the quantitative metrics and the qualitative evidence clearly distinguish the method from prior work. The current qualitative results do not give me confidence that the method achieves reasonable performance or that it generally outperforms existing approaches. Dexonomy demonstrates what is physically plausible functional grasp synthesis -- The presented results should at least match this level.

The paper also lacks real-world experiments.

For detailed reasoning regarding why the qualitative results are worrying, please see my comment “Clarification on main concerns.”

The rebuttal commits to include results on a broader set of objects and also argues that the current focus is on the simulation while the real-world exps are to be explored in future work. However:

- Given the current results on cup, sofa, and shoe objects, I am not confident that performance on other objects will be satisfactory.
- Real-world results may differ a lot from those in the simulation. I cannot believe that the method can achieve good performance in the real world.

For these reasons, I do not believe the paper meets the bar for acceptance at this time.

**Limitations:**

- Quality. The quality of the results is not satisfactory. Moreover, the paper lacks a thorough discussion and analysis of the experimental outcomes, which weakens the overall impact and clarity of the contributions.
- Method and the presentation. The methodological description and overall presentation are unclear. It is difficult to grasp the detailed process proposed in the paper, which raises doubts about whether the method can effectively address the grasping pose *translation* problem.

**Quality:**

2

**Strengths And Weaknesses:**

Strengths

- The paper presents a well-formulated problem and an interesting approach. Framing grasp pose translation as a stochastic process that transforms the source distribution into the target distribution is both a strong and reasonable formulation. The use of SB to model the relationship between the two distributions, combined with OT to complete the pose transformation, is well-designed and conceptually sound.
- The inclusion of algorithms in the method section enhances clarity and facilitates quick understanding of the approach.

Weaknesses

- **Experiments**: The experimental results raise several concerns regarding quality. As shown in Figure 3 and the supplementary figures, although the generated grasp poses do not exhibit significant penetrations into the object, they appear physically unstable. Furthermore, the qualitative results are limited and insufficient to convincingly demonstrate the method’s effectiveness, as only results on two objects—*sofa* and *cup*—are included. In particular, the generated poses for the *cup* differ significantly from the source poses, suggesting that the method sacrifices too much visual similarity in favor of physical validity.

  Additionally, no real-world experiments are presented. This absence raises further doubts about whether the generated poses are truly physically plausible in practical settings.

- **Presentation**: The paper's presentation requires substantial improvement. In Section 4, it is unclear why equations from Section 3 are reused without further clarification. In Section 4.2, the integration of the proposed cost terms into the OT process is not adequately explained. Simply stating "following XXX" is insufficient; the authors should not assume that readers are fully familiar with the referenced works. Clear and self-contained explanations are essential for understanding the methodology.

---

> ### Author Rebuttal · Authors · 2025-07-31
>
> We sincerely thank the reviewer for their time and insightful feedback. We are encouraged that the reviewer found our approach "well-formulated" and "interesting." We address each point below.
>
> > *W1*: On Experimental Quality, Plausibility, and Analysis
>
> **Physical Plausibility and Validation**
>
> We acknowledge that static images can sometimes appear unstable due to challenging viewpoints or occlusion. However, we want to emphasize that all generated grasps are quantitatively validated for physical stability in the high-fidelity IsaacGym simulator. We followed the rigorous protocol from GenDexGrasp [1], which we detail on Lines 292 - 299 in our manuscript:
>
> "We report the grasp success rate using IsaacGym, following the GenDexGrasp [1] evaluation protocol. A grasp is considered successful if it maintains a stable hold on the object under six perturbation trials, where external forces are applied along $\pm x$, $\pm y$, and $\pm z$ axes. We apply each force as a uniform acceleration of 0.5 m/s$^2$ for 60 simulation steps and measure whether the object translates more than 2 cm. A grasp passes if it withstands all six trials. Simulation parameters include a friction coefficient of 10 and an object density of 10,000. We use IsaacGym’s built-in positional controller to achieve the desired joint configurations."
>
> Our experimental results show that our method generates grasps that are not only stable but are **more stable** than all baselines. In response to other reviewers' feedback, we have also compared against two additional baselines, Dex-Retargeting [2] and CrossDex [3]. Our method continues to achieve the highest success rate, as shown below.
>
> ***Table 1: Comparison on Success Rate (%)↑***
> |                      | H→A   | H→S   | S→A   | Mean  |
> |----------------------|-------|-------|-------|-------|
> | Dex-Retargeting      | 56.93 | 16.21 | 56.44 | 43.19 |
> | CrossDex Retargeting | 36.51 | 8.97  | 35.12 | 26.87 |
> | RobotFingerPrint     | 66.80 | 33.89 | 70.31 | 57.00 |
> | Ours (Contact OT)    | ***74.78*** | ***37.10*** | ***76.37*** | ***62.75*** |
> | Ours (GWH OT)        | **77.73** | **42.59** | **78.74** | **66.34** |
>
> **Qualitative Results and Visual Similarity**
>
> We accept the criticism that the qualitative results in the initial submission were limited to two objects. Unfortunately, we are not allowed to present any multimedia results via external links as per the rebuttal guideline. We promise to include a more diverse set of qualitative examples from our 34 unseen test objects in the final version.
>
> Regarding the "generated poses for the cup differ significantly from the source poses," we wish to reiterate the core goal of our work. Our formulation of **grasp translation** is object-centric and aims to transfer the **functional intent** of a grasp, which allows for morphologically and posturally different but functionally equivalent outcomes.
>
> For example, to turn on a TV remote, a source grasp might be from below, palm up, with the thumb on the power button. A teleoperation system would try to replicate this exact pose. Our method, however, recognizes the functional goal (applying force to the power button while supporting the remote) and could generate a target grasp from above, palm down, with the index finger on the button and other fingers providing support. We consider both grasps **functionally equivalent** though they might appear very different visually.
>
> We believe the qualitative examples for the cup in Figure 3 all share a similar functional intent: the hand approaches from the bottom, with fingers touching the bottom of the cup, and the thumb positioned near the handle for stability. Our method successfully transfers this intent, while the baseline method RobotFingerPrint [4] fails to generate a stable grasp for this specific example.
>
> > *W2*: Regarding real-world experiments.
>
> We agree that real-world validation is crucial. Given the short rebuttal period, we were unable to conduct these experiments. Our work currently focuses on the core algorithmic contributions, validated in simulation. We are committed to including real-world experiments in future work.
>
> However, to strengthen the paper now, we have conducted extensive new simulation experiments, including comparisons to two additional strong baselines (Dex-Retargeting [2], CrossDex [3]), evaluation with a new pose alignment metric, a detailed efficiency report, and evaluation on the remaining three transfer settings (A→H, S→H, and A→S). Please refer to Table 1-5 of our response to reviewer AEMy and Table 2-4 of our response to reviewer 4u79 if interested. We believe these additions strengthen our empirical results and address key questions about performance and applicability.
>
> > *W3 & L2*: On Methodological Clarity
>
> Thank you for pointing out areas where the presentation can be improved. We will revise the manuscript to be more self-contained. To clarify the link between Sections 3 and 4, we will add the following at the beginning of Section 4.1:
>
> "We now apply the general Schrödinger Bridge framework from Section 3 to our specific problem of grasp translation. The abstract distributions $q_0$ and $q_1$  become the latent distributions of source and target grasps, respectively. The transport process between them is learned by optimizing the SF$^2$M objective from Eq. 7 in a latent space."
>
> To clarify the integration of OT costs, we will add the following at the beginning of Section 4.2:
>
> "To guide the Schrödinger Bridge, we must first compute an entropic OT plan $\pi_{\varepsilon}^* $ (Eq. 2), which is defined by a ground cost $d(\cdot, \cdot)$. We design several physics-informed costs for this purpose. For a given cost, we compute the OT plan between minibatches of source and target latent samples. We then draw pairs $(z_0, z_1) \sim \pi_{\varepsilon}^*$ and use these pairs to construct the conditional training targets for the SB model as described in Section 3.2 and used in the loss function of Eq. 7."
>
> We will also move some of the general formulation details from Section 3 to the appendix to improve the main manuscript's flow. To provide the intuition the reviewer requested, we will add a new figure to Sec 4.1 showing the learned transport process on a 2D toy example (e.g., translating from a two-moons distribution to an 8-Gaussians distribution). This will visually illustrate the latent evolution concept that underpins our method.
>
> > *Q1 & Q2*: Role of Grasp Pose $g_s$ and Encoding Input
>
> This is a great catch. The $o_\mathrm{hand}$ in Figure 2 represents the visual observation of the hand, which is used as the input to the VAE encoder. The grasp configuration $g$ is used during VAE training as part of the reconstruction loss in Eq. 8, but it is not an input to the SB model. In Algorithms 2 and 3 (SB training and inference), $g_s$ is not used. The process is conditioned only on the encoded visual observation $o_s$. We will correct the algorithms to remove $g_s$ from the requirements for Latent SB training and inference to avoid confusion.
>
> > *Q3*: Metric on SO(3).
>
> We apologize for the lack of clarity. We use the squared Frobenius norm between the two rotation matrices in SO(3). The 6D continuous representation is first converted to a $3\times 3$ matrix representation via Gram-Schmidt orthogonalization. The term is actually $\left\lVert R_{\mathrm{source}} - R_{\mathrm{target}} \right\rVert^2_F$ in Eq. 9. We will add this explicit definition to the paper.
>
> > *Q4*: Are the generated grasping poses validated in a physics simulator?
>
> Yes. As detailed in our response to W1, all grasps are rigorously tested for stability in the IsaacGym physics simulator using a 6-axis perturbation protocol.
>
> > *L1*: Thorough discussion and analysis of the experimental outcomes.
>
> We agree that a more thorough analysis would strengthen the paper. We will add the following discussion to Section 5 to better interpret the results from Table 1:
>
> "Our results in Table 1 reveal a key trade-off between the different physics-informed objectives. The Jacobian-based cost yields the highest mean success rate (67.45%), suggesting it is most effective at capturing the kinematic relationship for producing stable grasps. However, the GWH and Contact costs achieve superior functional alignment, measured by 6D GWH IoU (14.63% and 14.54%, respectively). This indicates that directly optimizing for force-exertion capabilities (GWH) or contact patterns (Contact) is more effective for transferring the grasp's underlying mechanical intent, even if this sometimes results in a slightly lower stability score compared to a pure manipulability-based objective (Jacobian)."
>
> #### Reference:
> [1] Li et al., "Gendexgrasp: Generalizable dexterous grasping." ICRA 2023.
>
> [2] Qin et al., "Anyteleop: A general vision-based dexterous robot arm-hand teleoperation system." RSS 2023.
>
> [3] Yuan et al., "Cross-embodiment dexterous grasping with reinforcement learning." ICLR 2025.
>
> [4] Khargonkar et al., "RobotFingerPrint: Unified Gripper Coordinate Space for Multi-Gripper Grasp Synthesis and Transfer." arXiv 2024.

---

> > ### Comment · Reviewer_Xi3Y · 2025-08-05
> >
> > Thanks for the detailed rebuttal. After reading the authors' clarifications and the comments from other reviewers, my main concerns regarding the limited evaluations and constrained quality of the work remain. Therefore, I will maintain my original rating.

---

> > > ### Author Response · Authors · 2025-08-06
> > > **Post-rebuttal discussion**
> > >
> > > We thank the reviewer for the follow-up and for specifying their remaining concerns. In our detailed initial rebuttal, we believe we answered all technical questions and proposed specific revisions to improve the manuscript's clarity and analytical depth. We will now focus on the two remaining concerns.
> > >
> > > **On "Limited Evaluations"**
> > >
> > > We respectfully argue that our evaluation is comprehensive. In the rebuttal, we added two key baselines, a detailed efficiency report, and results on all six transfer settings. Although showing additional qualitative examples is limited by rebuttal guidelines, the two objects selected in the manuscript are representative of our tests on **34 unseen objects** with greatly varying shapes, sizes, and visual appearances. We promise to incorporate a more diverse set of qualitative examples.
> > >
> > > Crucially, our method achieves **the highest success rate in physics simulation**, which provides objective proof of stability. We also highlight two points that strengthen our evaluation's relevance:
> > > 1. Our method only requires **vision-based input**, which is more practical for real-world scenarios than the state-based inputs (e.g., joint configurations, object poses) required by all baselines.
> > > 2. Our goal is to transfer **functional intent**, not visual mimicry, which explains visual differences in poses. Our design choice is supported by our superior performance on functional metrics such as the 6D GWH IoU and the new Contact Alignment metric, which measures the squared Chamfer distance between source and target grasp contact maps, as shown below.
> > >
> > > ***Table 1: Comparison on 6D GWH IoU (%)↑***
> > > |                      | H→A   | H→S   | S→A   | Mean  |
> > > |----------------------|-------|-------|-------|-------|
> > > | Dex-Retargeting      | 5.67  | 4.41  | 6.97  | 5.68  |
> > > | CrossDex Retargeting | 5.69  | 4.24  | 5.86  | 5.26  |
> > > | RobotFingerPrint     | 7.51  | 6.62  | 8.99  | 7.77  |
> > > | Ours (Contact OT)    | **15.89** | **15.18** | ***12.54*** | ***14.54*** |
> > > | Ours (GWH OT)        | ***15.09*** | ***14.14*** | **14.67** | **14.63** |
> > >
> > > ***Table 2: Comparison on Contact Alignment (cm^2)↓***
> > > |                      | H→A   | H→S   | S→A   | Mean  |
> > > |----------------------|-------|-------|-------|-------|
> > > | Dex-Retargeting      | ***9.53***  | 12.92 | ***6.68***  | 9.71  |
> > > | CrossDex Retargeting | 9.85  | 14.52 | 7.06  | 10.48 |
> > > | RobotFingerPrint     | 19.21 | 16.66 | 11.27 | 15.71 |
> > > | Ours (Contact OT)    | **5.05**  | **3.37**  | **6.54**  | **4.99**  |
> > > | Ours (GWH OT)        | 10.71 | ***7.26***  | 10.19 | ***9.39***  |
> > >
> > > **On "Constrained Quality"**
> > >
> > > We acknowledge the need to improve the paper's presentation. As detailed in our rebuttal response, we will significantly improve the manuscript by:
> > > 1. **Improving Methodological Clarity:** We will restructure the paper to better explain our core contributions.
> > > 2. **Deepening the Analysis:** We will add a more thorough discussion interpreting the experimental outcomes.
> > >
> > > Given the new quantitative evidence of our method's high quality and our concrete plan to improve the paper's presentation, we would be very grateful if the reviewer could clarify **what specific evidence or revision** they feel is still needed for addressing "limited evaluations" and "constrained quality". We believe we have addressed the points from the initial review and are keen to understand what we may have missed.

---

> > > ### Comment · Area_Chair_eZ11 · 2025-08-06
> > > **Clarification**
> > >
> > > Dear Xi3Y, Could you please elaborate on what concerns remain regarding the limited evaluation, and whether the follow-up rebuttal has addressed any of them? If there are still unclear points, do you think the authors should clarify them further? A clearer understanding would help me better evaluate both the paper and the authors' response. Thank you!
> > >
> > > Your AC.

---

> ### Comment · Reviewer_Xi3Y · 2025-08-09
> **Clarification on main concerns**
>
> Dear Authors and AC,
>
> Thank you to the authors for the additional clarifications and to the AC for the follow-up questions. I also apologize for not articulating my concerns clearly earlier. I will lay out my concerns more clearly, taking the original paper, the rebuttal and clarifications, the authors’ stated plans into consideration.
>
> Transfer quality and fidelity
>
> - Functionality not convincingly preserved & Balancing functionality and visual similarity: While I understand that the method prioritizes preserving functionality when transferring grasps, the transferred poses often diverge substantially from the source. In Figure 1, the source grasp contacts the lid, body, and bottom of the bottle; the transferred grasp contacts only the lid and body, and the global hand orientation changes markedly. In Figure 4 (supp), the grasp pose of the third object shifts from one end to the other end. -- This degree of deviation harms not only visual similarity, but also functional similarity. -- How to use a hammer if the transferred pose has shifted from grasping its handle to grasping its head?
> - Value of transfer vs. generic synthesis: If the goal is solely to achieve functionality, a generic grasp-synthesis method that produces physically valid grasps with functionality guidance or filtering might suffice. A transfer method should preserve at least coarse visual/affordance similarity to the source.
> - Comparison to Dexonomy (RSS 2025; https://pku-epic.github.io/Dexonomy/): It synthesizes physically plausible grasps for diverse objects from human-demonstrated templates. Although it is not a transfer method per se, it seems capable of addressing the Grasp2Grasp problem by treating the source grasp as a template. Its results appear functionally and visually similar to the original grasps and physically plausible, whereas the Grasp2Grasp results lag in these aspects.
>
> Physical plausibility
>
> - The Shadow Hand example in Figure 1 appears physically unstable.
>
> Limited evaluations
>
> - Real-world evaluation: Real-world experiments are important and non-trivial. I cannot base my assessment on a promise to add them in the camera-ready version.
>
> Metrics
>
> - I am not convinced by the superiority claimed on the contact alignment metric. As seen in Figure 4 (supp.) and Figure 1, the contact regions shift substantially.
>
> I appreciate the additional evaluations conducted during the rebuttal period. However, I remain unconvinced that the transfer quality sufficiently improves over prior work. Accordingly, I cannot raise my rating.

---

> > ### Author Response · Authors · 2025-08-09
> > **Discussion on remaining concerns**
> >
> > Thank you for the detailed follow-up. We hope to resolve your remaining concerns with the clarifications below.
> >
> > **Functionality vs. Visual Similarity**
> >
> > We respectfully disagree that the demonstrated grasps show a harmful deviation in functionality. We believe some concerns here may stem from misinterpretations of the 2D visualizations.
> > - **Figure 1 (Bottle):** We apologize for this confusing example. The bottle is largely rotationally symmetric around its minor axis. With the background floor pattern as a reference, the source and translated grasps share similar global hand orientations. We will release an interactive viewer that locks the object frame to remove viewpoint ambiguity. While the reviewer note a difference in the contact pattern (lid/body vs. lid/body/bottom), we have inspected the grasps in a 3D viewer and can confirm that subtle but important functional contacts exist. For instance, the Allegro hand's ring finger contacts the lower body for stability, and the Shadow hand's pinky finger (occluded in the image) makes contact with the side, forming a stable grasp where the baseline fails entirely.
> > - **Figure 4 (Shoe):** We respectfully believe the claim that the grasp "shifts from one end to the other" is a misinterpretation. The source grasp approaches from the toe. Our Contact OT variant reproduces this pattern very closely, and our other variants keep parallel palm alignment and bilateral side contacts. In contrast, although the RFP baseline may appear visually closer in global pose, it results in severe interpenetration and is not physically valid.
> > - **Objective evidence of functional preservation:** Beyond contact similarity, our method also considers functional similarity with GWS overlap, and Jacobian-based manipulability. Quantitatively, we outperform baselines in all functional alignment and physical stability metrics, indicating that mechanical capability is preserved and grasps remain stable. The contact alignment metric, which is object-frame canonical, further demonstrates this objectively.
> >
> > **Transfer vs. Generic Synthesis**
> >
> > We completely agree with your premise: "a generic grasp-synthesis... might suffice." The critical question is **how to provide the functionality guidance**. Our contribution is to use **a single visual observation of the source grasp as the task specification**, then translate it to another embodiment with correspondence automatically discovered by our OT design, **without needing paired demonstrations, annotated affordance maps, or human-specified grasp types**, which is a novel and more practical problem formulation.
> >
> > **Comparison to Dexonomy (RSS 2025)**
> >
> > We note that Dexonomy is **concurrent work** under NeurIPS policy, as it appeared online after the March 1st, 2025 cutoff. Therefore, a comparison is not expected. Nonetheless, the **problem assumptions differ**: Dexonomy's "Type-Conditional Model" requires a pre-defined "grasp type" from a codebook at inference and human-annotated contacts/normals templates for its optimization-based transfer. In contrast, our method requires **no grasp types**, **no human-labeled contacts**, learns from **unpaired visual observations**, and only needs **a single source point cloud at inference**. The claim that our results "lag" behind is not supported by comparable empirical evidence.
> >
> > **On Physical Plausibility**
> >
> > Regarding the example in Figure 1, this is primarily due to occlusion. We inspected the exact scene in a 3D viewer: the pinky indeed contacts the body, forming a pinch. More importantly, the source grasp itself is fragile. Our method **remedies this instability** by generating a more robust grasp, demonstrating the ability to transfer and improve upon the functional intent of a suboptimal source grasp.
> >
> > **On Real-World Evaluation**
> >
> > We agree that real-world validation is important and have committed to it for future work. However, we believe that for a venue like NeurIPS, core algorithmic contributions validated thoroughly in simulation are valuable, and the lack of physical experiments should not be the sole reason for rejection.
> >
> > **On Metrics**
> >
> > Regarding the contact alignment metric, we maintain that the perceived "substantial shift" in contact regions is an artifact of the 2D viewpoint, while the contact alignment metric itself is object-frame canonical. Our quantitative results show that our method achieves significantly better contact alignment than all baselines, which often fail to make physically plausible contact at all. We believe the functional and coarse visual similarity will be explicit with 3D visualizations.
> >
> > Regarding the comment that the reviewer is "unconvinced that the transfer quality sufficiently improves over prior work," we have demonstrated superiority over all comparable baselines across multiple metrics measuring stability, diversity, and functional similarity. If this comment refers to Dexonomy, we reiterate that it is concurrent work with different, stronger assumptions.

---

### Official Review · Reviewer_DVx1 · 2025-07-01

**Clarity:** 2
**Significance:** 3
**Originality:** 3
**Rating:** 4
**Confidence:** 3

**Summary:**

The paper introduces a vision-based dexterous grasp translation method framed as a Schrödinger Bridge problem. This method aims to translate grasp intentions from a source robotic hand to a target hand with different morphologies without relying on paired demonstrations. By employing physics-informed cost functions and learning distributional grasp translations, the method attempts to bridge morphological gaps between diverse robotic manipulators.

**Questions:**

- Regarding the human hand scenario, the input to the VAE consists of rendered images of a skeletal hand model, making practical application to actual human hands infeasible. This effectively treats the human hand as merely another robotic hand model. Could the authors clarify how their approach can be realistically extended to accommodate real human hand data?
- It is unclear whether the input to the VAE is RGB image or point cloud. Figure 2 indicates point cloud, while the text description indicates RGB. If it’s RGB, how to do the segmentation in real-world? What if the input image is a highly occluded one, which is quite common in dex mainpulation? If it’s point cloud, how to obtain it during inference?
- Do we need contact anchor during inference? My current understanding is “yes”. But how can we get it during inference?

**Ethical Concerns:**

["NO or VERY MINOR ethics concerns only"]

**Final Justification:**

Most of my questions have been answered in the rebuttal, and I have accordingly raised my rating. However, there are still several reasons why I am unable to give a higher score:

1. The responses highlight a key limitation: the current pipeline is difficult to deploy in real-world scenarios. It relies on oracle contact anchors and clean, segmented point clouds as input — both of which are extremely challenging to obtain in real-world inference, especially in dexterous grasping tasks where occlusions are common.

2. The writing of the paper remains quite rough. While the authors have proposed some revisions in the rebuttal and promised to improve the clarity, the extent of revision required — including adjustments to some core claims — is substantial. Given this, I am not confident that all necessary changes can be adequately addressed in the camera-ready version. It would be best to undergo another round of revision to fully address these issues.

**Limitations:**

See Weakness and Question

**Quality:**

2

**Strengths And Weaknesses:**

**Strengths:**

- The proposed method leverages an innovative Schrödinger Bridge formulation, enabling a principled probabilistic approach to grasp translation.
- Introducing physics-informed cost functions is a valuable contribution, effectively guiding the grasp translation process to maintain functional grasp similarity despite morphological differences.
- The method demonstrates promising performance in simulation environments across multiple hand-object pairs, indicating strong potential for generalization.

**Weaknesses:**

- The paper appears to be slightly over-claiming, which needs further clarification:
    - A central claim made by the authors is that *“While recent works in data-driven grasp generation have shown success in generating feasible grasps for specific hand object pairs, they often rely on large-scale datasets curated through computationally expensive simulation or real-world trials.”.* However, the proposed method itself is trained on MultiGripperGrasp, a substantial dataset containing approximately 30.4 million grasps across 11 robotic manipulators and 345 objects. Thus, the claim is misleading.
    - More critically, another prominent claim is about overcoming the hand-specific limitations of existing models *“Moreover, such models typically operate in a hand-specific setting, which requires retraining or significant fine-tuning to adapt to new robotic hands.”*. The authors argue their approach generalizes to unseen dexterous hands without requiring retraining or significant fine-tuning. To validate this claim meaningfully, the authors need explicit evidence demonstrating robust generalization to unseen robotic hands not included in the training dataset. Otherwise, the claimed advantage over existing hand-specific methods becomes questionable.
- Tables and figures are not self-contained or sufficiently explained. For instance, in Figure 3, terms like "pose" and "contact" lack clear definitions within the figure caption, causing confusion and reducing readability.
- The paper lacks real-world experiments entirely. Without empirical validation in real-world scenarios, the practicality and applicability of the proposed method remain questionable.

---

> ### Author Rebuttal · Authors · 2025-07-31
>
> We sincerely thank the reviewer for their time and insightful feedback. We are encouraged that the reviewer found our formulation "innovative" and our contribution "valuable." We address each point below.
>
> > *W1 & W2*: On Claims, Datasets, and Generalization
>
> We thank the reviewer for highlighting a potential ambiguity in our paper's motivation versus its direct claims. We agree that the introduction could be framed more precisely, and we will revise it accordingly.
>
> **Our Core Contribution: Translation without *Paired* Data**
>
> The central motivation of our work is to overcome the need for paired demonstration data for grasp translation. While single-hand grasp datasets can be generated efficiently with modern simulators (e.g., GraspIt! [1], GenDexGrasp [2]), creating a large-scale dataset of corresponding *functionally equivalent* grasps across two different hands is extremely expensive and often intractable.
>
> Our method addresses this bottleneck. We use the MultiGripperGrasp dataset [3] for its rich collection of single-hand grasps, but our framework **does not use or require any pre-existing pairings** between them. Instead, the Schrödinger Bridge formulation, guided by our physics-informed OT designs, learns to discover these functional correspondences automatically from unpaired sets of grasps. This is analogous to machine translation: while monolingual text data is abundant, high-quality parallel corpora (paired sentences) are scarce and expensive. Our work provides a new perspective on how to scale up grasp translation by leveraging readily available, unpaired grasp examples.
>
> **Clarification on Unseen Hand Generalization**
>
> We also wish to clarify that generalizing to unseen hands is a long-term **motivation** for this research area, not a **claimed contribution** of our current model. We believe our approach is a meaningful step towards this motivation. The current implementation does not generalize to unseen hands at inference time because our VAE decoder learns the hand's kinematic structure implicitly from the training data. To generate a grasp for a new hand, the decoder would need to be retrained or fine-tuned on data from that hand. We acknowledge this limitation and view the development of learning-based methods that can generalize to new hand kinematics on-the-fly as an important and open research direction.
>
> To address these points and prevent misunderstanding, we will revise the introductory section (Lines 16-22) as follows:
>
> "A key challenge in data-driven grasp synthesis is the heavy reliance on large-scale datasets, which are often curated for a specific hand through computationally expensive simulation or real-world trials. Transferring grasp knowledge to a new robotic hand is non-trivial, as models trained for one hand often require complete retraining or significant fine-tuning for another, which poses a practical bottleneck. A particularly challenging aspect is the acquisition of **paired demonstration data**, where a grasp for a source hand is mapped to a corresponding functional grasp for a target hand. Manually creating or algorithmically discovering such pairings at scale is extremely difficult. This motivates the need for a framework that can learn to translate the *intent* of a grasp from a source hand to a target hand using only unpaired, hand-specific grasp datasets while leveraging the underlying object geometry and physics to bridge the morphological gap."
>
> > *W3*: Tables and figures are not self-contained.
>
> Thank you for this feedback. We will improve the clarity of our figures. For Figure 3, the labels "Pose," "Contact," "GWH," and "Jacobian" refer to variants of our model trained with the different OT cost functions described in Section 4.2. We will add the following sentence to the caption to make it self-contained:
>
> "The 'Pose', 'Contact', 'GWH', and 'Jacobian' columns show results from our method when trained using the respective OT cost functions from Sec. 4.2."
>
> We will perform another round of proofreading on all figures and tables to ensure they are self-contained and easy to understand.
>
> > *W4*: Lack real-world experiments
>
> We agree that real-world validation is a crucial step. Given the rebuttal timeframe, we were unable to conduct and document physical experiments. Our current research has focused on the core algorithmic contributions and their validation in a high-fidelity simulator. We fully recognize the importance of this aspect and are committed to including real-world experiments in future work.
>
> However, to further strengthen the paper's evaluation based on the reviewers' feedback, we have conducted extensive new simulation experiments, including comparisons to two additional strong baselines (Dex-Retargeting [4], CrossDex [5]), evaluation with a new pose alignment metric, a detailed efficiency report, and evaluation on the remaining three transfer settings (A→H, S→H, and A→S). Please refer to Table 1-5 of our response to reviewer AEMy and Table 2-4 of our response to reviewer 4u79 if interested. We believe these additions strengthen our empirical results and address key questions about performance and applicability.
>
> > *Q1*: Human hand as source and extension to real human data.
>
> This is an excellent question. The human hand in our experiments is a fully articulated 3D model derived from MANO [6]. Specifically, we use distinct part meshes for each link, which are kinematically chained. While this means the complete mesh surface can appear discontinuous at the joints, this has little negative impact on our method. The input to our VAE is a **point cloud** sampled from the surfaces of these parts, which is an inherently discrete representation. Thus, the visual difference between a point cloud sampled from our articulated model versus one from a continuous, skinned MANO model is not significant.
>
> Therefore, our method can be directly applied to real human hand data, assuming that a system can provide a full, segmented point cloud of the hand grasping the object. We will provide some thoughts on how to obtain that in our response to your next question.
>
> > *Q2*: Input modality to VAE
>
> We apologize for the confusion. The input to the VAE is a **full hand point cloud**, as shown in the architecture diagram (Figure 2). We will clarify this in the text.
>
> We acknowledge that obtaining a complete, segmented point cloud in the real world is challenging due to occlusion. This is indeed a common problem in robotic perception. Standard solutions include:
> 1. **Multi-camera systems** to fuse views and reconstruct a more complete point cloud, which is often seen in real-world experimental setups.
> 2. **Pose estimation**: If 3D models of the object and hand are known, the problem can be reduced to a 3D hand-object pose estimation (HOPE) task, which is generally well studied with 3D inputs [7]. Once the poses are estimated, complete point clouds can be generated from the models and used as input to our system.
>
> > *Q3*: Regarding contact anchor
>
> Yes, your understanding is correct. Contact anchors are required at inference. They are derived directly from the assumed input: the segmented source hand and object point clouds. The procedure is deterministic and computationally inexpensive:
> 1. For each point on the object, we calculate its minimum distance to the source hand point cloud.
> 2. Points with a distance below a small threshold (e.g., 0.005m in our experiments) are identified as the contact map on the object.
> 3. We then apply Farthest Point Sampling (FPS) to this contact map to select the 5 contact anchors.
>
> This process requires no additional information beyond the input point clouds that are already conditioned upon. We will clarify this procedure in the implementation details.
>
> #### Reference:
> [1] Miller and Allen. "Graspit! a versatile simulator for robotic grasping." RA-M 2004.
>
> [2] Li et al., "Gendexgrasp: Generalizable dexterous grasping." ICRA 2023.
>
> [3] Casas et al., "Multigrippergrasp: A dataset for robotic grasping from parallel jaw grippers to dexterous hands." IROS 2024.
>
> [4] Qin et al., "Anyteleop: A general vision-based dexterous robot arm-hand teleoperation system." RSS 2023.
>
> [5] Yuan et al., "Cross-embodiment dexterous grasping with reinforcement learning." ICLR 2025.
>
> [6] Romero et al., "Embodied hands: Modeling and capturing hands and bodies together." SIGGRAPH ASIA 2017.
>
> [7] Hampali et al., "Honnotate: A method for 3d annotation of hand and object poses." CVPR 2020.

---

> > ### Comment · Reviewer_DVx1 · 2025-08-06
> >
> > Thank you to the authors for their clarification. Most of my questions have been answered, and I have accordingly raised my rating. However, there are still several reasons why I am unable to give a higher score:
> >
> > 1. The responses highlight a key limitation: the current pipeline is difficult to deploy in real-world scenarios. It relies on oracle contact anchors and clean, segmented point clouds as input — both of which are extremely challenging to obtain in real-world inference, especially in dexterous grasping tasks where occlusions are common.
> >
> > 2. The writing of the paper remains quite rough. While the authors have proposed some revisions in the rebuttal and promised to improve the clarity, the extent of revision required — including adjustments to some core claims — is substantial. Given this, I am not confident that all necessary changes can be adequately addressed in the camera-ready version.

---

> > > ### Author Response · Authors · 2025-08-06
> > > **Post-rebuttal response**
> > >
> > > We sincerely thank the reviewer for engaging with our rebuttal and for raising their ratings. We appreciate the opportunity to address the final remaining concerns.
> > > 1. **On Real-World Deployment:** We agree that the reliance on clean, segmented point clouds is a key limitation for deployment. However, we would like to point out that our framework is fundamentally designed around **vision-based inputs**. This is in contrast to the state-based inputs (e.g., joint configurations, object pose) required by other baselines, which are often less accessible in practice. While obtaining clean visual data can be challenging, we believe our method's reliance on vision is still a meaningful step towards practical real-world deployments.
> > > 2. **On Writing and Revisions:** We want to reaffirm our commitment to delivering a high-quality camera-ready version. We have a clear and detailed plan to implement all the proposed changes as detailed in our rebuttal. We are confident that these revisions are manageable and will result in a more polished and clear manuscript.

---

### Official Review · Reviewer_AEMy · 2025-07-03

**Clarity:** 3
**Significance:** 2
**Originality:** 3
**Rating:** 4
**Confidence:** 4

**Summary:**

This paper presents an approach to grasp translation across robot hands. It formulates the problem as a stochastic transport between grasp distributions of different hands and learns to map between two latent distributions via score and flow matching. The translation is guided using alignment cost functions, such as pose, contact maps, wrench space, and manipulability alignment. Experiments include comparisons with previous methods and a diffusion baseline in terms of quality, diversity, and alignment, as well as ablation studies for diffusion rate and discretization steps.

**Questions:**

See weaknesses.

**Ethical Concerns:**

["NO or VERY MINOR ethics concerns only"]

**Final Justification:**

Thanks for the new baselines and the profiling results. Since real experiments and the pose metric are still missing, the reviewer will keep the original score.

**Limitations:**

Yes.

**Quality:**

3

**Strengths And Weaknesses:**

**Strengths:**

1. The formulation of stochastic transport for grasp translation and learning the mapping via score and flow matching is novel and interesting.
2. Compared to previous works, the four types of alignment (pose, contact maps, wrench space, and manipulability alignment) are comprehensive.
3. The proposed method outperforms the previous method RFP in all metrics in Table 1.
4. The presentation quality is high, and the figures are clear.

**Weaknesses:**

1. It is unclear which alignment metric is most suitable for the task. The paper adopts 6D GWH IoU as the alignment metric, and the contact map alignment cost achieves better alignment. However, this metric may not be useful for teleoperation tasks, which are likely the main application of grasp translation. It would be better to incorporate pose alignment metrics for evaluation.
2. Since grasp translation is often used for real-time teleoperation, a detailed efficiency report is expected, especially for the diffusion-based methods.
3. While the main contribution is latent score and flow matching for grasp translation, it would be better to include more baselines such as Dex-Retargeting, which is widely used in teleoperation papers, or neural retargeting method like CrossDex. The reviewer would like to see comparisons in terms of success rate, pose alignment, and efficiency.
4. Real-world experiments would make the paper more convincing.

Dex-Retargeting: https://github.com/dexsuite/dex-retargeting
CrossDex: https://github.com/PKU-RL/CrossDex/

---

> ### Author Rebuttal · Authors · 2025-07-31
>
> We sincerely thank the reviewer for their time and insightful feedback. We are encouraged that the reviewer found our formulation "novel and interesting" and our presentation "of high quality." We address the identified weaknesses below.
>
> > *W1-3*: Regarding pose alignment metric, efficiency report, and comparison with Dex-Retargeting and CrossDex.
>
> We appreciate the reviewer's suggestions regarding evaluation metrics, efficiency, and comparisons to other relevant work. We group these points as they all relate to the application context of our method.
>
> **Clarification: Grasp Translation vs. Teleoperation**
>
> While our work is technically related to teleoperation, we wish to clarify that our formulation of *grasp translation* has a fundamentally different goal.
> - *Teleoperation* is typically manipulator-oriented, often relying on a human-in-the-loop for perception and feedback, and aims to mimic the source agent's actions as closely as possible.
> - Our formulation of *grasp translation* is object-centric and aims to transfer the functional intent of a grasp, which does not necessarily require a human source and allows for morphologically and posturally different but functionally equivalent outcomes.
>
> For example, to turn on a TV remote, a source grasp might be from below, palm up, with the thumb on the power button. A teleoperation system would try to replicate this exact pose. Our method, however, recognizes the functional goal (applying force to the power button while supporting the remote) and could generate a target grasp from above, palm down, with the index finger on the button and other fingers providing support. We consider both grasps *functionally equivalent*.
>
> Back to your question about metrics, in our problem setting, we can imagine that two functionally equivalent grasps are likely to induce similar contact patterns (like in the remote example, contact at the bottom to fight gravity and at the on/off button for functionality) and/or wrench profiles (force and torque the grasp can exert). Similar contact patterns often lead to similar wrenches, as one can only exert force during contact, but similar wrenches do not necessarily indicate a similar contact pattern. You can imagine that flipping a water bottle requires torques about any axis on the horizontal plane, where any pinches on the side of the object are likely to induce these types of torques.
>
> This distinction motivates our choice of the 6D Grasp Wrench Hull (GWH) IoU as a primary alignment metric. Functionally equivalent grasps should be able to exert similar forces and torques on the object, which is precisely what GWH measures.
>
> **Additional Experiments: Baselines and Metrics**
>
> As per the reviewer's suggestion, we have conducted new experiments to include the positional retargeting module from Dex-Retargeting [1] as baselines. We also tweak the retargeting neural network from CrossDex [2] to be trained with generated data from positional retargeting to output both wrist pose and finger joint configuration as an additional baseline. Further, we introduce a new Contact Alignment metric as a proxy to pose alignment metric, inspired by Dex-Retargeting's positional optimization objective, which measures the squared Chamfer distance between source and target grasp contact maps.
>
> *A note on baselines*: Both Dex-Retargeting and CrossDex require pre-defined link-to-link correspondences between the source and target hands. This information is not typically available in large-scale grasp datasets, and we had to manually annotate it for these comparisons. This requirement limits their generality. For instance, they cannot retarget to a hand with more fingers than the source. Our method has no such constraints.
>
> The results, shown in the tables below, demonstrate that our method significantly outperforms these new baselines across all key metrics.
>
> ***Table 1: Comparison on Success Rate (%)↑***
> |                      | H→A   | H→S   | S→A   | Mean  |
> |----------------------|-------|-------|-------|-------|
> | Dex-Retargeting      | 56.93 | 16.21 | 56.44 | 43.19 |
> | CrossDex Retargeting | 36.51 | 8.97  | 35.12 | 26.87 |
> | RobotFingerPrint     | 66.80 | 33.89 | 70.31 | 57.00 |
> | Ours (Contact OT)    | ***74.78*** | ***37.10*** | ***76.37*** | ***62.75*** |
> | Ours (GWH OT)        | **77.73** | **42.59** | **78.74** | **66.34** |
>
> ***Table 2: Comparison on Diversity (rad)↑***
> |                      | H→A   | H→S   | S→A   | Mean  |
> |----------------------|-------|-------|-------|-------|
> | Dex-Retargeting      | 0.221 | 0.186 | 0.196 | 0.201 |
> | CrossDex Retargeting | 0.195 | 0.201 | 0.223 | 0.206 |
> | RobotFingerPrint     | 0.225 | 0.186 | 0.199 | 0.203 |
> | Ours (Contact OT)    | **0.294** | **0.211** | **0.311** | **0.272** |
> | Ours (GWH OT)        | ***0.293*** | ***0.197*** | ***0.309*** | ***0.266*** |
>
> ***Table 3: Comparison on 6D GWH IoU (%)↑***
> |                      | H→A   | H→S   | S→A   | Mean  |
> |----------------------|-------|-------|-------|-------|
> | Dex-Retargeting      | 5.67  | 4.41  | 6.97  | 5.68  |
> | CrossDex Retargeting | 5.69  | 4.24  | 5.86  | 5.26  |
> | RobotFingerPrint     | 7.51  | 6.62  | 8.99  | 7.77  |
> | Ours (Contact OT)    | **15.89** | **15.18** | ***12.54*** | ***14.54*** |
> | Ours (GWH OT)        | ***15.09*** | ***14.14*** | **14.67** | **14.63** |
>
> ***Table 4: Comparison on Contact Alignment (cm^2)↓***
> |                      | H→A   | H→S   | S→A   | Mean  |
> |----------------------|-------|-------|-------|-------|
> | Dex-Retargeting      | ***9.53***  | 12.92 | ***6.68***  | 9.71  |
> | CrossDex Retargeting | 9.85  | 14.52 | 7.06  | 10.48 |
> | RobotFingerPrint     | 19.21 | 16.66 | 11.27 | 15.71 |
> | Ours (Contact OT)    | **5.05**  | **3.37**  | **6.54**  | **4.99**  |
> | Ours (GWH OT)        | 10.71 | ***7.26***  | 10.19 | ***9.39***  |
>
> Our analysis suggests the baselines struggle because their IK-style optimization focuses only on select distal links, ignoring potential collisions from other parts of the hand (e.g., the palm), which leads to physically invalid grasps. The CrossDex model, trained on this suboptimal data, inherits these limitations and fails to generalize due to its simple architecture and small size (4-layer MLP). We will integrate these results into the final paper and add this discussion to the related works.
>
> **Efficiency Report**
>
> We provide a more detailed inference time report below. While our method is designed primarily as an offline planner, its amortized inference time is on the same order of magnitude as the baselines and significantly faster than the optimization-based RobotFingerPrint [3]. CrossDex Retargeting network is significantly faster due to its simple architecture, and it only needs to do the forward pass once.
>
> ***Table 5: Average Inference Time per Grasp, Amortized (s)***
> |                       | H→A    | H→S    | S→A    |
> |-----------------------|--------|--------|--------|
> | Dex-Retargeting       | 0.13   | 0.16   | 7.98e-2 |
> | CrossDex Retargeting  | 1.1e-5 | 1.1e-5 | 1.1e-5 |
> | RobotFingerPrint      | 5.2    | 5.8    | 5.1    |
> | Diffusion (100 steps) | 0.5    | 0.5    | 0.5    |
> | Ours (100 steps)      | 0.8    | 0.8    | 0.8    |
>
> > *W4*: Regarding real-world experiments.
>
> We agree that real-world validation is a crucial step to demonstrate the full impact of our work. Given the short rebuttal period, we were unable to conduct these experiments. However, our current work focuses on the core algorithmic contributions for grasp translation, which we believe are sound and well-supported by our extensive simulation results. We are fully committed to incorporating real-world experiments in future work.
>
> #### Reference:
> [1] Qin et al., "Anyteleop: A general vision-based dexterous robot arm-hand teleoperation system." RSS 2023.
>
> [2] Yuan et al., "Cross-embodiment dexterous grasping with reinforcement learning." ICLR 2025.
>
> [3] Khargonkar et al., "RobotFingerPrint: Unified Gripper Coordinate Space for Multi-Gripper Grasp Synthesis and Transfer." arXiv 2024.

---

> > ### Comment · Reviewer_AEMy · 2025-08-06
> >
> > ## Grasp translation VS Teleopeartion
> >
> > Thanks for the clarification. However, I still find it hard to believe that contact maps or GWH are more functionally aligned than fingertip positions.
> >
> > Both contact maps and GWH are still indirect metrics. They do not represent the actual force or torque applied to the object after simulation. A grasp with high GWH IoU may still fail to produce the intended force. In some cases, a grasp that is closer in position (e.g., fingertip pose) could result in even more similar forces after simulation.
> >
> > For teleoperation, although position alignment may not always result in identical forces, it offers higher controllability. If there is no directly measuring of the final force (e.g., through simulation), it is still reasonable to consider pose alignment as a valuable metric.
> > ## New baselines and profiles
> >
> > Thanks for the new baselines and the profiling results. They make the claims more convincing. One more question: while the translation is done in an offline fashion, what is the main reason it is slow? Is it just due to the model size?
> >
> > ## Clarification
> >
> > Thanks for the new baselines and the profiling results. Since real experiments and the pose metric are still missing, the reviewer will keep the original score.

---

> > > ### Author Response · Authors · 2025-08-06
> > > **Post-rebuttal response**
> > >
> > > Thank you for the thoughtful follow-up and for acknowledging our new experiments. We appreciate the opportunity to clarify these final points.
> > >
> > > **On Alignment Metrics**
> > >
> > > We agree that all proxy metrics (e.g., GWH, contact maps) are indirect measures of the final interaction forces. Our motivation for using GWH and contact maps is that they capture the necessary **pre-conditions** for applying force, which is critical when translating between hands with different morphologies. A target hand with a different palm shape or finger length may require a different wrist pose to achieve the same functional contact pattern, making direct fingertip-to-fingertip alignment potentially misleading or kinematically infeasible.
> > >
> > > Nevertheless, we take the reviewer's suggestion on exploring metrics based on final exerted forces/torques seriously for future work. The primary challenge, which led us to use these pre-condition metrics, is that successful static grasps are in force-closure, where the net force/torque is zero. The actual exerted force is then dependent on external perturbations (e.g., gravity), making it difficult to define a canonical "functional force" for comparison. We believe our current metric set provides a holistic validation, but we agree this is a valuable direction.
> > >
> > > **On Inference Time**
> > >
> > > Regarding inference time, there are two main factors contributing to its length:
> > > 1. **Model Size:** As the reviewer suggests, model size is a factor. Our U-ViT based architecture (\~45M parameters) is substantially larger than lightweight baselines like CrossDex's 4-layer MLP (\~500k parameters).
> > > 2. **Iterative Process:** More fundamentally, the time is due to the **iterative sampling process** inherent to diffusion and flow-based generative models. To solve the Schrödinger Bridge dynamics, we perform 100 sequential integration steps with Euler–Maruyama, with each step requiring a full forward pass. This contrasts sharply with methods like CrossDex that require only a single forward pass.

---

### Note · Authors · 2025-08-11

We thank all reviewers for their time and invaluable feedback. Our work introduces a novel Schrödinger Bridge framework for vision-based grasp translation across different hand morphologies. Guided by physics-informed Optimal Transport plans, our method learns to transfer functional grasp intent from unpaired data, requiring only a single source point cloud at inference.

We are encouraged that reviewers found our core formulation "novel and interesting" [AEMy], "innovative" [DVx1], "well-formulated" [Xi3Y], and "insightful" [4u79]. The physics-informed costs were recognized as a "valuable contribution" [DVx1], and our experiments were noted for demonstrating "substantial improvements" over prior work [4u79]. We are grateful for this positive reception of our main ideas.

To address the concerns raised, we have performed a significant amount of new work. To strengthen our evaluation [AEMy, Xi3Y], we added two key baselines, a new contact alignment metric, a detailed efficiency report, and results for all six transfer settings. Our method demonstrates superior performance across all quantitative metrics.

In the final version, we will incorporate all promised revisions. To improve clarity and address concerns about our claims [DVx1, Xi3Y], we will revise the introduction to precisely frame our contribution around learning from unpaired data and restructure the methodology to highlight our core technical details better. We will also add more qualitative examples and 2D/3D visualizations to provide intuitions as suggested [Xi3Y, 4u79].

We are confident that the revised manuscript will be a clear, well-supported, and significant contribution. Thank you for your consideration.

---

### Decision · Program_Chairs · 2025-09-17

**Decision:**

Accept (poster)

**Comment:**

The paper presents a novel cross-morphology grasp transfer method based on Schrödinger Bridges. Reviewers found the approach both novel and probabilistically principled, though initial concerns were raised about evaluation, clarity, and qualitative results.

Post-rebuttal, these concerns were largely addressed with stronger evaluations, baselines, and metrics. The main remaining issue was the lack of real-world experiments. The AC agrees with the authors that NeurIPS primarily rewards algorithmic contributions, and given the paper does not claim system-level robotics contributions, thorough simulation studies are sufficient. Reviewer Xi3Y also noted contact stability issues and requested a comparison with Dextonomy, but since that work is concurrent (later acknowledged by Reviewer), the AC excluded it from consideration. In comparison to prior baselines, the proposed method already demonstrates superior performance, while some improvement space does remain, as pointed out by Xi3Y.

Overall, the AC finds the method and evaluation sufficient to justify acceptance, and encourages the authors to include the promised revisions and 3D interactive visualizations to better highlight grasping quality.